# The effect of rainfall amount and timing on annual transpiration in a grazed savanna grassland

Matti Räsänen[1], Mika Aurela[2], Ville Vakkari[2,3], Johan P. Beukes[3], Juha-Pekka Tuovinen[2], Pieter G. Van Zyl[3], Miroslav Josipovic[3], Stefan J. Siebert[4], Tuomas Laurila[2], Markku Kulmala[1], Lauri Laakso[2,3], Janne Rinne[5], Ram Oren[6,7,8], and Gabriel Katul[6,9]

[1]Institute for Atmospheric and Earth System Research, University of Helsinki, Finland
[2]Finnish Meteorological Institute, Helsinki, Finland
[3]Atmospheric Chemistry Research Group, Chemical Resource Beneficiation, North-West University, Potchefstroom, South Africa
[4]Unit for Environmental Sciences and Management, North-West University, Potchefstroom, South Africa
[5]Department of Physical Geography and Ecosystem Science, Lund University, Sweden
[6]Nicholas School of the Environment, Duke University, Durham, North Carolina, USA
[7]Department of Forest Science, University of Helsinki, Finland
[8]Pratt School of Engineering, Duke University, Durham, North Carolina, USA
[9]Department of Civil and Environmental Engineering, Duke University, Durham, North Carolina, USA

*Correspondence to*: Matti Räsänen (matti.rasanen@helsinki.fi)

## Abstract

The role of precipitation ($P$) variability on evapotranspiration (ET) and its two components, transpiration ($T$) and evaporation ($E$) from savannas, continues to draw significant research interest given its relevance to a number of eco-hydrological applications. Our study reports on six years of measured ET and estimated $T$ and $E$ from a grazed savanna grassland in Welgegund, South Africa. Annual $P$ varied significantly in amount (508 to 672 mm yr$^{-1}$), with dry years characterized by infrequent early-season rainfall. $T$ was determined using annual water-use efficiency and gross primary production estimates derived from eddy covariance measurements of latent heat flux and net ecosystem $CO_2$ exchange rates. The computed annual $T$ for the four wet years with frequent early wet-season rainfall was nearly constant, $326 \pm 19$ mm yr$^{-1}$ ($T$/ET=0.51), but was lower and more variable between the two dry years (255 and 154 mm yr$^{-1}$). Annual $T$ and $T$/ET were linearly related to the early wet-season storm frequency. The constancy of annual $T$ during wet years is explained by the moderate water stress of $C_4$ grasses, and trees' ability to use water from deeper layers. During extreme drought, grasses respond to water availability with a dieback-regrowth pattern, reducing leaf area and transpiration, increasing the proportion of transpiration contributed by trees. The works suggest that the early-season $P$ distribution explains the interannual variability in $T$, which should be considered in managing the grazing and fodder production at these grasslands.

# 1    Introduction

Similar to other semi-arid areas, wooded grasslands in central South Africa deliver essential ecosystem services such as grazing land and fodder (Bengtsson et al., 2019). In such semi-arid zones, evapotranspiration (ET) approximately matches annual precipitation ($P \approx 500$ mm yr$^{-1}$; Zhang et al., 2001). The transpiration ($T$) component accounts for water loss from the leaf stomata of the sparse tree component, seasonal grasses, and the minor forb component. The evaporation ($E$) component is large following rain events, as intercepted water and near-surface soil water evaporate; the latter may continue over periods longer than a week (Perez-Priego et al., 2018). The partition of ET between $E$ and $T$ may affect the net radiation ($R_n$) and surface temperature on short timescales (sub-daily). However, the processes that increase the proportion (and amount) of water used in $T$, facilitating greater carbon uptake and subsequent fodder production for cattle, occur over timescales of weeks or longer. Given the link between $T$ and carbon uptake from the atmosphere, there is growing interest in how ET is partitioned into $E$ and $T$ in semi-arid ecosystems (Merbold et al., 2009; Sankaran et al., 2004; Scanlon et al., 2002, 2005; Scholes and Archer, 1997; Scott and Biederman, 2017; Volder et al., 2013; Williams and Albertson, 2004; Xu et al., 2015; Yu and D'Odorico, 2015). The aim here is to explore this partition of ET using a long-term data set of measured fluxes of energy, water, carbon dioxide ($CO_2$), and vegetation activity from a grazed wooded grassland. The focus is restricted to processes operating over timescales ranging from daily to seasonal, commensurate with controls over the annual partition of $P$ into $T$, and the resulting carbon uptake in gross primary production (GPP). These longer timescales are of interest in the valuation of ecosystem productivity and their services when assessing climatic shifts (Godde et al., 2020). The results presented here on the partitioning of ET must be viewed as necessary but insufficient for developing best practices for the management of grazing or fodder production.

The contrasting vegetation layers of wooded grasslands have distinct seasonal dynamics of leaf area and physiological activity. In southern African drylands, *Vachellia erioloba* (Camel-thorn tree) is an important woody species (Barnes et al., 1997), which is a deep-rooted semi-deciduous tree with a low leaf turnover rate, resulting in minor leaf area changes. Furthermore, this species has been shown to absorb 37% of its water below a depth of 1 m, partially decoupling its physiological activity from recent precipitation and shallow soil water content (Beyer et al., 2018). Little interannual variation in tree water use has been shown in many semi-arid ecosystems (Do et al., 2008; Hutley et al., 2001; Montaldo et al., 2020). However, pronounced seasonality in tree transpiration of semi-deciduous and deciduous savanna tree species has been observed in South Africa, even at riparian forest (Scott-Shaw and Everson, 2019). Perennial $C_4$ grass species in the area have shallower rooting systems and are physiologically responsive to intermittent rainfall events (Sankaran, 2019). The grass dependence on the temporal distribution of $P$ has been demonstrated by a positive correlation between rainfall frequency and productivity at field scale (Swemmer et al., 2007) and by a positive relationship between wet season rainfall frequency and grass cover in the

grassland ecosystems of sub-Saharan Africa ($P \leq 630$ mm yr$^{-1}$; D'Onofrio et al., 2019). Compared to $C_3$ trees, the $C_4$ grass layer has $CO_2$ concentrated in the bundle sheath that enables greater light and water-use efficiencies of $CO_2$ uptake in the warmer intercanopy spaces (Ripley et al., 2010). Due to their ability to regulate intercellular $CO_2$ concentrations, $C_4$ grasses have higher photosynthesis per unit leaf area that can be sustained even in moderate water stress situations (Taylor et al., 2014). However,

for the same volume of soil, a $C_4$ grass is an intensive and fast user of soil water when compared to $C_3$ trees. Indeed, due to their shallow rooting depth, severe droughts may alter both their water-use efficiency per unit leaf area and their leaf area dynamics. Thus, our study objective is not only to partition measured ET into $T$ and $E$ but also to quantify the effect of environmental variables on the seasonality of the grass activity.

Three methods that link $T$ to GPP are used to estimate monthly $T$/ET. These methods were chosen because previous applications showed some success in partitioning ET into $E$ and $T$ when applied to multi-site data sets. These three methods provide an estimate of ecosystem-scale $T$, albeit with differing assumptions and uncertainty (Stoy et al., 2019). Previous method comparisons have shown that the linear regression method with optimality assumption produces lower $T$/ET estimates than the machine learning approach to ET partitioning for grassland and savanna ecosystems (Nelson et al. 2020; Scott et al.,

2021). Comparing these methods allows selecting the most suitable partitioning scheme for water-limited ecosystems in general and savannas in particular. Furthermore, an agreement between the methods lends confidence to the estimates of $T$/ET and the drivers of $T$ (e.g., precipitation). Disagreements between the methods may also identify potential uncertainties for the hydroclimatic or land-cover conditions explored here. Hence, a corollary goal is to understand the opportunities and limitations of these methods when combined with eddy covariance-measured ET such as those supplied by FluxNet (Baldocchi et al.,

20    2001).

The main question addressed here is how $T$ and $T$/ET vary with $P$ at monthly and annual timescales in a grazed savanna grassland ecosystem. Available MODIS vegetation indexes and Landsat 8 grass and tree Normalized Difference Vegetation Index (NDVI) allow quantifying vegetation dynamics at the site. Soil moisture-based grass transpiration and ecosystem scale

$T$ offer a new perspective on the relation between water fluxes at seasonal and annual timescales, and ways to examine the role of grass and trees in water budgets. Our study objectives are (i) to quantify the variation in annual $P$, ET, and $T$, (ii) identify the main drivers of the annual and monthly $T$ and $T$/ET, and (iii) relate the growth dynamics of tree and grass components to the hydrological balance.

## 2    Materials and methods

### 2.1    Site description

The Welgegund measurement site is located in a grazed savanna grassland in South Africa (26°34′10″S, 26°56′2″E, 1480 m.a.s.l.), shown in Figure 1. The research site is part of a large-scale commercial ranch with an annual cattle head count of 1300 ± 300. The cattle grazing area is approximately 6000 ha.

The area experiences two seasonal periods: a warm rainy season from October to April and a cool dry season from May to September. The 16-year mean annual rainfall determined at a nearby weather station (town of Potchefstroom) was 540 mm yr$^{-1}$ ±112 mm yr$^{-1}$ (Räsänen et al., 2017). The soil around the site is loamy sand in the top 1 m. Although the water table depth is not known, the farm well has a continuous water supply at 30 m below the surface (Fig. 1).

The vegetation in the area is an open thornveld. *Eragrostis trichophora*, *Panicum maximum*, and *Setaria sphacelata* are the dominant perennial C$_4$ grass species. The mean maximum grass height across sampling plots was 0.1 m in 2011 (Räsänen et al., 2017). Tree cover is 15%, and the dominant tree species is *Vachellia erioloba,* with other less prominent species such as *Celtis africana* and *Searsia pyroides*. *Dicoma tomentosa*, *Hermannia depressa*, *Pentzia globosa*, and *Selago densiflora* are the dominant forb species. Details about the site and vegetation cover may be found elsewhere (Jaars et al., 2016; Räsänen et al., 2017).

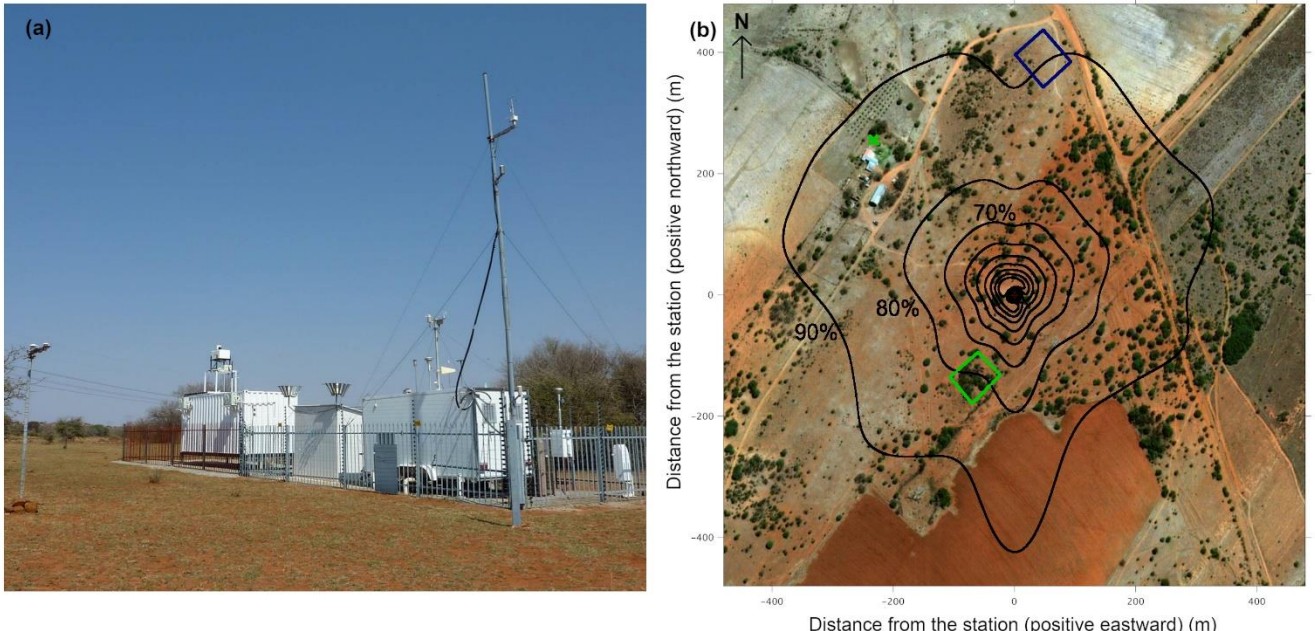

**Figure 1. (a) Photograph of the measurement station and surrounding vegetation taken on 22 August 2016. (b) Daytime flux footprint estimated with measured flow statistics and a 2D footprint model (Kljun et al., 2015). Footprint contour lines (black) are shown in 10% increments from 10 to 90%. The station is located at the center, and the green cross indicates the farm well. The Landsat 8**
**NDVI was calculated from the area marked by the blue square for grass and by the green square for trees.**

## 2.2   Measurements

Atmospheric aerosols, trace gases, and meteorological variables were measured continuously at the site (Beukes et al., 2015; Petäjä et al., 2013). The measurements directly related to energy fluxes and water balance are briefly described. The eddy covariance (EC) system consisted of a triaxial sonic anemometer (METEK USA-1) and a Li-Cor (LI-7000) closed path infrared

gas analyzer, which were positioned 9 m above the ground surface. The sampling frequency of the EC system was 10 Hz. The gas analyzer was calibrated every month with a high-precision $CO_2$ span gas using synthetic air with $CO_2 < 0.5$ ppm as a reference gas. The meteorological measurements included atmospheric temperature and pressure, mean wind speed and direction, and mean air relative humidity. The radiation measurements were made using Kipp & Zonen PAR-lite sensors, CMP-3 pyranometers, and a NR-lite2 net radiometer positioned 3 m above the ground with a field of view at the grass level.

These sensors measure photosynthetically active radiation (PAR), direct and reflected global radiation, and net radiation. The soil surface heat flux was measured with a Hukseflux HFP01 heat flux plate at 5 cm below the soil surface. The meteorological variables were sampled every 1 minute (radiation every 10 seconds), and 15 min averages were then recorded.

Precipitation was measured 1.5 m aboveground with two tipping bucket gauges (Vaisala QMR102 and Casella). Most of the

precipitation values reported here are from the Vaisala gauge, with the Casella gauge only used to gap-fill missing values in

the Vaisala gauge time series. No significant differences were observed between the two sensors. Between December 2011 and February 2012, the measured rainfall was underestimated due to the high intensity of the rainfall, and it was corrected using nearby weather station measurements (Sect. S.1, Fig. S1 and S2). Wind-induced underestimation is a well-known problem with pointwise rainfall measurements. Thus, the measured precipitation was corrected by multiplying the measured

precipitation by 1.094. This corresponds to the 9.4% bias that was determined for the Casella rain gauge at the height of 0.5 m at a measurement site with a similar mean wind speed (5 m s$^{-1}$) and annual rainfall ($P$=700–1000 mm yr$^{-1}$) (Pollock et al., 2018).

Stored soil water changes were determined using two separate soil moisture profiles. The measurements of individual soil
moisture sensors at depths 0.05, 0.2, and 0.5 m (Delta-T ML2) were converted to a single average soil moisture value using the weights of 125, 225, and 200 mm. These soil moisture measurements covered the complete experiment period from September 2010 to August 2016. Starting from March 2012 onwards, a Delta-T PR2/6 probe was installed to record soil moisture at 0.1, 0.2, 0.3, 0.4, 0.6, and 1.0 m depths. This profile measurement was converted to stored soil water using the weights 150, 100, 100, 150, 300, and 200 mm, and it was used to calculate the annual change in soil water storage ($\Delta\theta_{1m}$)
over the entire 1 m soil column.

The site was visited once or twice a week during the six-year period to check the status of the sensors and correct errors if necessary. Measurement records were used to identify anomalies, outliers, or erroneous measurement periods. Further details of the site and EC measurements are presented elsewhere (Aurela et al., 2009; Räsänen et al., 2017). The annual energy balance
closure was also computed, which varied from 0.75 to 0.85. This lack-of-closure is comparable to those reported in FluxNet sites (Stoy et al., 2013; Wilson et al., 2002). Given the heterogeneity in vegetation cover and that EC measurements sense a different footprint from the footprint representing the difference between net radiation and soil heat flux, no Bowen ratio adjustments were performed to force an energy balance closure.

## 2.3    Flux calculation and gap-filling

The details of the turbulent flux calculations are presented in Räsänen et al. (2017). Briefly, the turbulent fluxes were calculated as 30 min block averages after double rotation and by applying the Webb–Pearman–Leuning (WPL) density correction (Webb et al., 1980). The low-frequency flux correction was performed according to Moore (1986), and high-frequency losses were corrected using empirical transfer functions determined using sensible heat flux as a reference scalar. The sensible and latent heat flux values were discarded when the measured friction velocity $u_*$ was below 0.28 m s$^{-1}$, which was deemed as a state
of low turbulence mixing. The steady-state test of Foken and Wichura (1996) was used to screen the latent heat flux data for nonstationary conditions within each 30-min averaging period. The data were discarded if the relative nonstationarity defined by this test exceeded a threshold, which was set to 30% and 100% for the data used for gap-filling and final analysis, respectively. Latent heat fluxes were checked for an acceptable H$_2$O concentration range and variance to detect anomalous

spikes due to condensation or rainfall. Heat flux values were filtered for outliers by considering values for each month of all the measurement years and removing outliers using an adjusted boxplot (Hubert and Vandervieren, 2008). The steady-state check resulted in less than 30% filtered fluxes, which were gap-filled using marginal distribution sampling (MDS) from the REddyProc package (Reichstein et al., 2005). Both daytime and nighttime fluxes were gap-filled using this approach, given

the significant role nighttime evaporation and respiration play in the water and carbon balances. Gap-filling of nighttime evaporation is of significance at Welgegund, as most of the rainfall occurs in the late afternoon and early evening. The meteorological parameters were also gap-filled using the MDS approach (Reichstein et al., 2005).

The flux footprint was estimated using the daytime measured flow statistics for the six-year period and a standard 2D footprint

model (Kljun et al., 2015), which are presented in Fig. 1. These calculations suggest that 80% of the ET fluxes originate from the homogeneous thornveld.

The EC-inferred GPP was used to derive the water-use efficiencies to partition measured ET into $T$ and $E$. The measured net ecosystem $CO_2$ exchange (NEE) was partitioned into GPP and ecosystem respiration using nighttime mean respiration values.

These values were assumed to be the same for daytime respiration, and GPP was determined as the difference between NEE and daytime ecosystem respiration. Nighttime mean respiration was used instead of the exponential temperature function, as only 2% of the fitting windows had a linear or exponential relation between EC-based ecosystem respiration and soil temperature. The difference between the mean monthly transpiration from these two methods was small, with transpiration from the exponential temperature function being 4% higher than transpiration from the nighttime mean method (Fig. S3). The

GPP fit parameters and the nighttime mean respiration were calculated in a moving data window that was defined for each day with an initial length of six days. The moving window was expanded by up to 20 days if necessary, to include at least 50 measurement points. The measured NEE had one large 25-day gap in September 2013, and the fit parameters were linearly interpolated for this gap. The preprocessing of NEE was performed with the same filters as the heat fluxes, as discussed in Räsänen et al. (2017).

The potential ET (PET) was calculated using the Priestley–Taylor formulation given by Priestley and Taylor (1972)

$$PET = \alpha_{PT} \frac{\Delta}{\Delta + \gamma_p} (R_n - G), \tag{1}$$

where $\alpha_{PT} = 1.26$ is the Priestley–Taylor coefficient, $\Delta = \mathrm{d}e^*/\mathrm{d}T_a$ (Pa K$^{-1}$), $e^*$ (Pa) is the saturation vapor pressure given by the Clausius–Clayperon equation and evaluated at the measured air temperature $T_a$ (K), $R_n$ (W m$^{-2}$) is net radiation, $G$ (W m$^{-2}$) is soil heat flux, and $\gamma_p$ (Pa K$^{-1}$) is the psychrometric constant. The energy balance closure (EBC) slope was estimated for

each year by regressing all measured half-hourly values of $R_n$–$G$ against the sum of the measured latent and sensible heat fluxes for the same period.

## 2.4 Uncertainty of annual ET estimates

Friction velocity ($u^*$) threshold was estimated using a bootstrap technique from 200 artificial replicates of the data set (Wutzler et al., 2018). The mean $u^*$ estimate value for the whole data set was 0.28 m s$^{-1}$, with heat flux and NEE values being discarded when $u^*$ was lower than this limit. The 5th, 50th, and 95th percentiles of the estimates were 0.27, 0.29, and 0.32 m s$^{-1}$, respectively. The data set was $u^*$ filtered and gap-filled with these three $u^*$ limits. The annual $u^*$ uncertainty range was calculated for each k year as

$$E_{u*,k} = \frac{\mathrm{ET_{max,k}} - \mathrm{ET_{min,k}}}{\mathrm{ET_{median,k}}} \mathrm{ET_k}, \tag{2}$$

where $E_{u*,k}$ is the $u^*$ uncertainty for year $k$ and $\mathrm{ET_k}$ is the evapotranspiration for year $k$.

The MDS gap-filling algorithm estimates random error for each half-hour value based on the standard deviation of the observed latent heat flux with similar meteorological conditions in a moving window. The annual random error was estimated as root-mean squared error

$$E_{rand,k} = \sqrt{\sum_{i=1}^{n_k} \sigma_i^2}, \tag{3}$$

where $n_k$ is the number of 30 min periods in year $k$ and $\sigma_i$ is the standard deviation of latent heat flux from the MDS gap-filling algorithm. The total uncertainty of the annual ET was calculated by adding the random error and $u^*$ uncertainty in quadrature to yield

$$E_{tot,k} = \sqrt{E_{u*,k}^2 + E_{rand,k}^2}. \tag{4}$$

## 2.5 Rainfall interception

The total rainfall interception ($I_t$) was not measured but estimated by modeling grass, litter, and tree interception. The interception was estimated for each storm using discrete rainfall events separated by at least one hour. The grass interception for one storm event was calculated using an expression derived for crops (Moene and Van Dam, 2014)

$$I_{grass} = a\mathrm{LAI}\left(1 - \frac{1}{1 + \frac{c_g P_g}{a\mathrm{LAI}}}\right), \tag{5}$$

where $a$ is the scale parameter, $c_g$ is the grass cover fraction, LAI is leaf area index estimated here from satellite (Sect. 2.7), ,and $P_g$ is the rainfall amount per storm. The scale parameter was set to 0.5 mm (event)$^{-1}$, which corresponds to a maximal 1 mm interception loss for LAI=2. The grass cover fraction was estimated using LAI:

$$c_g = 1 - e^{-k\mathrm{LAI}}, \tag{6}$$

where the extinction coefficient ($k$) is set to 0.4. Tree interception was estimated using the revised model for a sparse canopy (Gash et al., 1995). The model assumes that rainfall events consist of wetting, saturation, and drying phases. The interception

for small events that do not saturate the canopy was estimated separately from large storms that saturate the canopy. The rainfall to fill the canopy storage is

$$P'_g = -S_c \left( \frac{\overline{R}}{\overline{E}} \right) \ln \left( 1 - \left( \frac{\overline{E}}{\overline{R}} \right) \right), \tag{7}$$

where $S_c = S/c_t$ is the canopy storage capacity per unit cover, $\overline{R}$ is the mean rainfall, and $\overline{E}$ is the mean evaporation rate during a storm. The measured ET was used to calculate the mean evaporation rate for each event. The tree cover fraction $c_t$ was set to a constant 0.15, and storage capacity $S$ was set to 1.07 mm, corresponding to a measured value for *A. mearnsii* (Bulcock and Jewitt, 2012). The total tree interception is then determined as

$$I_{tree} = c_t \sum_{i=1}^{m} P_{g,i} + \sum_{j=1}^{n} [ c_t P'_g + c_t \frac{\overline{E}}{\overline{R}} (P_{g,j} - P'_g)]. \tag{8}$$

The first sum accounts for the $m$ small events that do not saturate the canopy and the second sum accounts for the $n$ large events. The litter interception was assumed to be 1 mm per rainfall event (Scholes and Walker, 1993) and it was multiplied by $c_t$.

## 2.6 Partitioning ET

Prior to presenting the three ET partitioning approaches, the link between GPP and $T$ is reviewed. From definitions, the flux-based water-use efficiency (WUE) is:

$$\text{WUE} = \frac{\text{GPP}}{T} \propto \frac{c_a}{\text{VPD}} (1 - c_i/c_a), \tag{9}$$

where $c_i$ and $c_a$ are the intercellular and ambient atmospheric $CO_2$ concentrations and VPD is the vapor pressure deficit. Based on stomatal optimization theories that maximize carbon gain for a given amount of water loss in the rooting system per unit leaf area, the ratio of $CO_2$ concentrations $(1 - c_i/c_a)$ is proportional to $\sqrt{\text{VPD}}$, as demonstrated in several studies reviewed elsewhere (Hari et al., 2000; Katul et al., 2009, 2010). Combining these theories with the definition of WUE (Eq. 5) makes $T$ proportional to GPP $\times$ VPD$^{0.5}$ provided that $c_a$ does not vary appreciably. The proportionality constant in this expression ($T \propto$ GPP $\times$ VPD$^{0.5}$) is linked to the so-called marginal water-use efficiency (or the Lagrange multiplier in optimal stomatal control theories), which differs from the intrinsic water-use efficiency $iWUE = (1 - c_i/c_a)c_a$. It must be externally supplied or determined from EC measurements during conditions when $T$ approximately equals ET. When this proportionality constant is known, an EC-based GPP estimate (together with VPD) can be used to infer $T$ and, subtracting from ET, produce an estimate of $E$.

Three approaches were used to divide ET into $E$ and $T$ using the above-mentioned link between GPP and $T$ (Table 1). The first method was presented by Berkelhammer et al. (2016) (hereafter, B16). Here, it was applied to each year individually to allow for the large inter-annual variation in vegetation phenology. The method assumes that ET is linearly related to GPP $\times$ VPD$^{0.5}$

only when $T$ is the dominant term in ET. Also, the $T$/ET ratio is assumed to approach unity intermittently. To estimate the $T$/ET value for each 30-min period, the product GPP by $VPD^{1/2}$ was plotted against ET for each year, and the minimum value of ET was then selected as the fifth percentile for each equal-sized GPP $\times$ $VPD^{0.5}$ bin. The bin was defined by discretizing the 30 min GPP $\times$ $VPD^{0.5}$ values into 50 bins, each containing the same number of measurements, but encompassing different value ranges, for reasons provided elsewhere (Berkelhammer et al., 2016). A linear regression line of these bins defines the ET value for which $T$ dominates ET. Any value falling below the line is considered to have $T$/ET=1. For points above the regression line, $T$/ET is defined as the ratio between the minimum ET that represents $T$ and the observed ET:

$$\frac{T}{\text{ET}} = \frac{\min_{\text{GPP}}||\text{ET}||}{\text{ET}_{\text{flux}}}, \tag{10}$$

where $\min_{\text{GPP}}||\text{ET}||$ is the minimum ET value and $\text{ET}_{\text{flux}}$ is the observed ET value. The calculation of half-hour $T$/ET values for one year is illustrated in Figure S4. The daily $T$/ET values were calculated by taking the mean of these half-hour $T$/ET values, and it was used to calculate daily $T$ and $E$ using the measured ET. The regression slope and intercept of the $T$=ET line are related to the inverse of water-use efficiency for each year based on the half-hour data. The 30 min data points used for the $T$/ET estimation were also filtered with additional quality criteria, i.e. only data points with measured ET, positive GPP, and $R_n$ were used (see Zhou et al., 2016). However, rainy days were included in the estimation to capture the rainfall interception events measured by the EC system while maintaining the data-stationarity filter. Shortly after rain, water droplets remaining on the sonic anemometer transducers can block the detection of sound waves emitted and received, leading to anomalous vertical velocity and friction velocity measurements for these 30 min runs. However, as the sonic anemometer transducers are inclined, smooth, and have small surface area, they dry out faster than the leaves, thereby allowing the EC system to operate shortly after each rainfall event. At an annual scale, the estimated $I_t$ was used to calculate soil evaporation ($E_s$) by subtracting $I_t$ from $E$.

As previously mentioned, two other methods were also used to estimate $T$ to identify the method most appropriate to water-limited ecosystems. The second approach entailed fitting the $T$=ET line using quantile regression with zero-intercept for each year; the slopes of these fitted lines are termed uWUEp (Zhou et al., 2016) (hereafter, Z16). The apparent uWUE slopes (uWUEa) were defined for each day separately by fitting the half-hour ET values to GPP $\times$ $VPD^{0.5}$ values using linear regression with zero intercept. The daily $T$/ET value is the ratio of uWUEa slope and uWUEp of each year. The difference between the Berkelhammer method and the uWUE method is primarily in the process of fitting the $T$=ET line.

The third approach is a random forest regressor that first isolates the most likely periods when $T$ is equal to ET, after which it trains on GPP and $T$ relations during these periods to infer $T$ from measured GPP (Nelson et al., 2018) (hereafter, N18). A summary of these methods and their requirements is featured in Table 1 for convenience. In addition, an ET partition method that does not assume equality between $T$ and ET during any periods was tested but could not be applied at this site because the monthly multiyear correlations between ET and GPP were not significant (Fig. S5) (Scott and Biederman, 2017).

**Table 1. Summary of the ET partitioning methods applied to the 6-year data.**

| Name | Input variables | Method to calculate daily $T$/ET | References |
|------|-----------------|----------------------------------|------------|
| B16 | ET, GPP, and VPD | Daily $T$/ET is an average of half-hour $T$/ET daytime values that are estimated during the measured flux periods. | Berkelhammer et al. (2016) |
| Z16 | ET, GPP, and VPD | Daily $T$/ET is the ratio of the slope of daily fit of $GPP \times VPD^{0.5}$ vs. ET (uWUEa) divided by the slope from the quantile regression of each year's data (uWUEp). | Zhou et al., (2016), Zhou et al., (2018), and Hu et al., (2018) |
| N18 | ET, GPP, air temperature, VPD, precipitation, incoming shortwave radiation, and wind speed | $T$ and $E$ are estimated for all half-hour periods, and daily $T$/ET is calculated using the daily sum of $T$ and ET. | Nelson et al., (2018) |

## 2.7 Stage-2 soil evaporation

The estimated daily $E$ was assessed using the stage-2 soil evaporation theory each year during the early dry season. During the stage-2 conditions of soil evaporation, the evaporation is controlled by the soil moisture and soil physical properties (desorptivity) (Brutsaert and Chen, 1995; Hu and Lei, 2021). After the rainfall event, the cumulative daily $E$ can be expressed

10 as $D_e t_d^{1/2}$, where $D_e$ is the soil desorptivity to be determined and $t_d$ is the dry-down duration in days. By regressing cumulative daily $E$ inferred from the aforementioned partitioning methods upon $\sqrt{t_d}$ for a single dry-down period, the $D_e$ can be computed and compared to literature values. The rainfall events were chosen from the end of April onwards, and all rainfall events were higher than 10 mm. The dry-down periods varied from 12 to 30 days. The expected range of $D_e$ based on several experiments is about 3 to 6 mm d$^{-1/2}$ for sandy soils (Brutsaert and Chen, 1995).

## 2.8 Grass transpiration

The grass transpiration ($T_g$) was estimated from step-shaped diurnal changes in soil moisture for the last four years (Jackisch et al., 2020). The method estimates root water uptake from the daily change of root zone soil moisture, and it was applied to each soil moisture measurement from 10 to 60 cm. Total daily grass transpiration is the sum of daily changes in each soil moisture layer. The algorithm considers only periods when soil moisture measurement has characteristic root water uptake decline and excludes periods of percolation (Jackisch et al., 2020). Twelve percent of the estimated $T_g$ values exceeded daily ET and were replaced with the daily ET values. The daily grass transpiration was set to zero during the dry season defined by MODIS LAI value less than 0.3. The portion of soil evaporation was removed from the grass transpiration estimate by removing a constant 0.28 mm d$^{-1}$, which is the mean daily dry season evaporation from the B16 method, from the estimated daily grass transpiration. This constant value was used instead of the daily $E$ from ET partitioning methods because the EC estimated daily $E$ values were unrealistically high compared to the estimated grass transpiration. This is because the soil moisture measurements may not fully capture the soil evaporation of the EC footprint.

## 2.9 Satellite data

Changes in vegetation cover were quantified using the monthly average of MODIS 16-day EVI with 250 m spatial resolution (MOD13Q1, collection 6) (Didan, 2015). The monthly average of MODIS 8-day LAI (MOD15A2H, collection 6) with 500 m spatial resolution was used to relate monthly $T$/ET to LAI, comparing estimated $T$/ET to variations in vegetation phenology. The EVI signal is a ratio of spectral bands, whereas the LAI has corrected units of foliage area per ground area. For the last three years, Landsat 8 L2 16-day NDVI with 30 m spatial resolution was used to determine separate grass and tree NDVI at the EC footprint (Fig. 1). Cloud and cloud shadow affected Landsat pixels were removed using automated cloud cover identification (Braaten et al., 2015).

## 2.10 Rainy season timing and green-up dates

Rainy season length ($T_{wet}$) was estimated based on a climatological threshold of 5% of the mean annual rainfall (Guan et al., 2014). The start of the rainy season was defined as the day when cumulative rainfall of the hydrological year (September to August) reached the threshold value of 27 mm, which was based on the long-term mean annual rainfall (540 mm yr$^{-1}$). Similarly, the end of the rainy season was estimated as the first day when cumulative rainfall, starting backward from the end of the hydrological year (August), reached the same threshold value. Early wet-season (September to November) precipitation was characterized by estimating the mean daily rainfall statistics using the daily mean precipitation amount ($\alpha$) and daily mean storm frequency ($\lambda$). The daily mean precipitation amount was calculated as the mean precipitation of rainy days, while the mean storm frequency was calculated as the inverse of the mean time between rainy days.

The tree green-up date, estimated from the raw 16-day EVI time series (Archibald and Scholes, 2007), was defined as the day when the EVI signal was higher than the moving average of the previous four time steps at the beginning of the hydrological year, which is the time when the EVI time series experiences a sudden increase.

## 3    Results

Before addressing the study objectives, we first present the variability in precipitation and LAI (or EVI). Next, the outcomes of the three ET partitioning methods summarized in Table 1 are featured. The tree and grass dynamics are outlined analyzing the monthly variability in grass and tree NDVI, $T$ and $T_g$. Finally, the annual water balance components are presented. Hereafter, hydrological years are defined as the time period from September to August and are referred to by the year in which they began.

### 3.1    Site meteorology and ET partitioning

The early-season rainfall was frequent in all years except 2011 and 2015 (Fig. 2a). The tree green-up days and start of the rainy period were not linearly related (Table S1, $R^2 = 0.03$, $p = 0.753$): the earliest tree green-up date occurred in 2011, 72 days earlier than the start of the rainy season. The year 2015 was an extreme drought year in South Africa and it was characterized by the lowest early season rainfall frequency and by rainy season length that was nearly twice as long as in other years (Fig. 2a). Water entered to the deeper soil layers in wet years 2012 and 2013 but not in the wet year 2014 and in the drought year 2015 (Fig. 2b). There was a two-week dry spell at the end of January 2011 and another dry spell at the end of November 2015, which are visible through the low measured ET and soil moisture values (Fig. 2b–c). Grass experienced dieback and regrowth in 2015 shown by decreased GPP during the early wet season (Fig. 2d). This is also seen in EVI trend, which decreases after initial increase (Fig. 2e). This period was also characterized by high VPD.

The daily $T$/ET estimated from the N18 method was consistently higher than the $T$/ET from B16 and Z16 methods (Fig. 2f). The difference between N18 and other methods was most significant during the end of wet seasons and during the drought year. The $T$/ET estimated from Z16 was higher than $T$/ET from B16 during the dry seasons. In the B16 method, the annually fitted line between the variable GPP $\times$ VPD$^{0.5}$ and ET established the empirical link between GPP and $T$. The bin values of the variable GPP $\times$ VPD$^{0.5}$ were linearly related to the fifth percentile of measured ET (Fig. S6), with the largest scatter occurring during the drought year ($R^2 = 0.85$, in 2015). For all years, the mean surface soil moisture during $T$=ET instances was 0.1 m$^3$m$^{-3}$ or less (Table S1). The annual slope of the $T$=ET was related to the rainy season length, with the year 2011 falling below the 95% confidence interval of the mean (Fig. 3a). The slope represents the $T$=ET values, and only 67% of those values were from the rainy season in 2011, as opposed to 75–84% in the other years (Table S1). The greater slope value in 2011 means lower water-use efficiency. The slope and intercept of the $T$=ET line were also linearly related (Fig. 3b). Thus, when

most $T$=ET values are observed during the rainy season, it is possible to estimate the annual $T$=ET line based only on the rainy season length for the B16 method. The annually fitted uWUEp for the Z16 method was related to the sum of $R_n$ during the wet season (Fig. 3c).

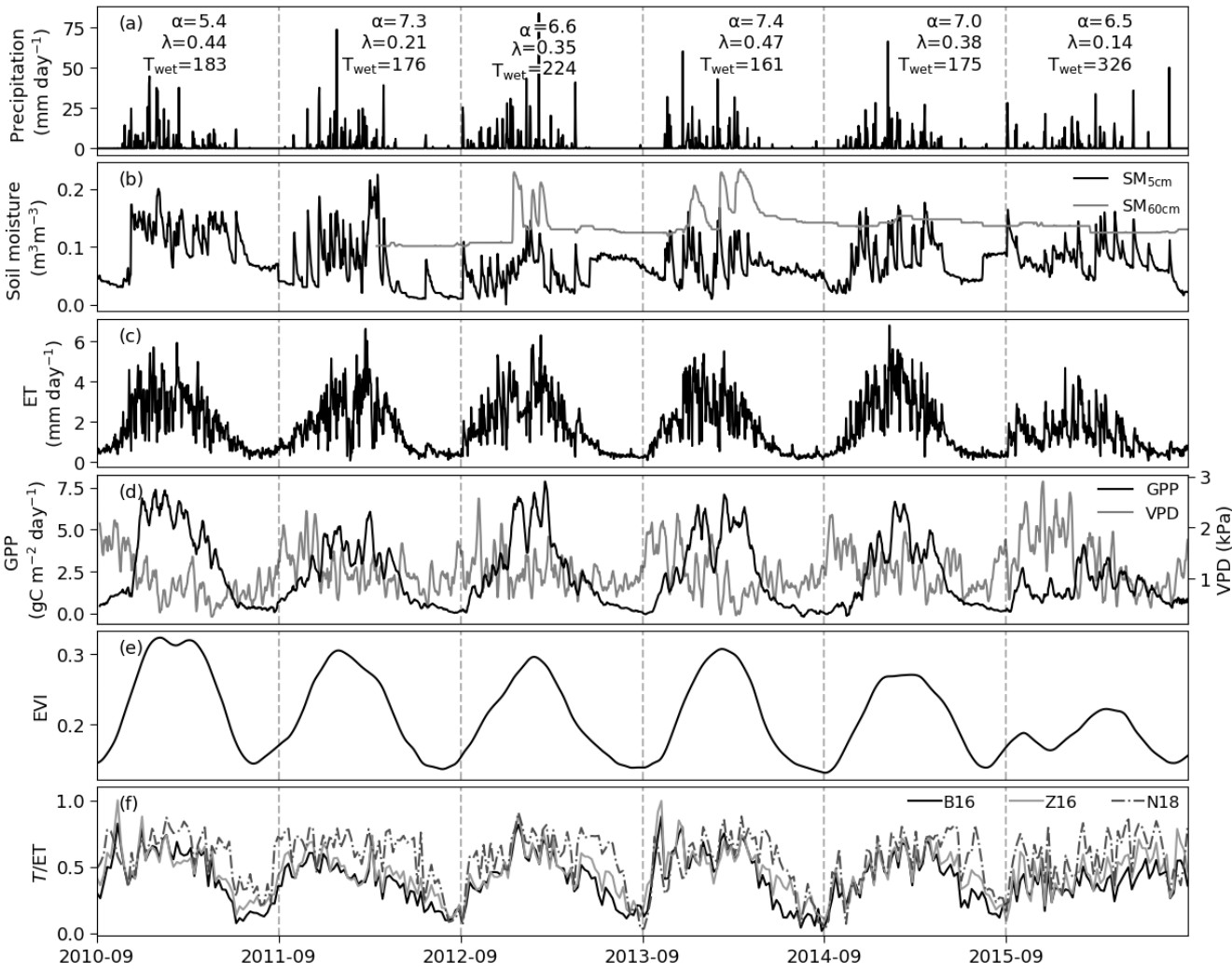

**Figure 2. Time series of daily precipitation, soil moisture at 10 and 60 cm depth, EC-based evapotranspiration, GPP, VPD, EVI, and 7-day average $T$/ET for B16, Z16 and N18 methods. The GPP and VPD lines show the 7-day running mean. Rainy season length $T_{wet}$ (days), daily mean precipitation amount ($\lambda$), and storm frequency ($\alpha$) for the early wet season (September to November) are indicated for each year in the top panel.**

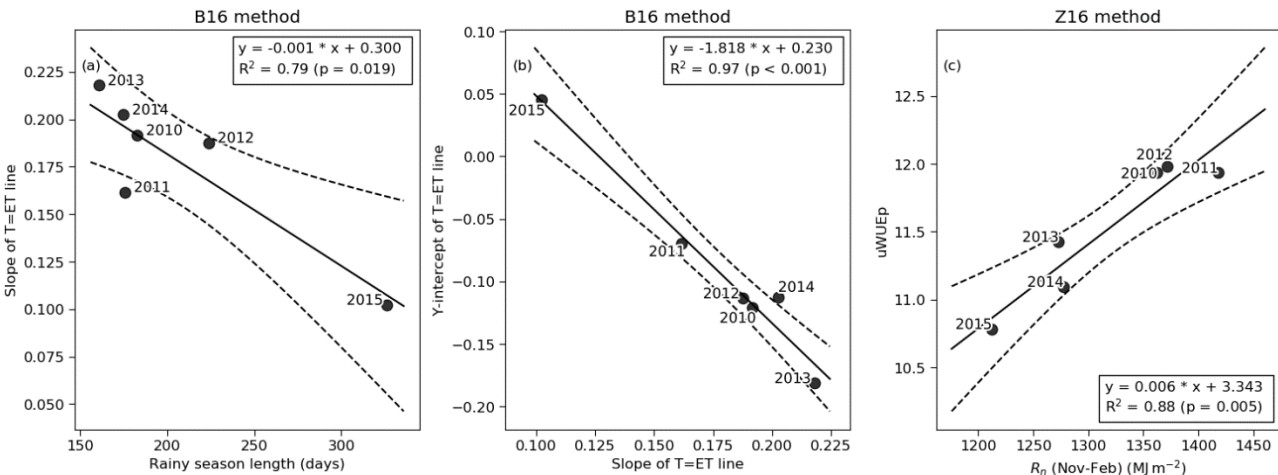

**Figure 3. (a) Relationship between the slope of T=ET line (B16 method) and rainy season length. (b) Relationship between the y-intercept and the slope of the T=ET line for the B16 method. (c) Relationship between annual uWUEp (Z16 method) and wet season $R_n$. Dashed lines demarcate the 95% confidence interval.**

The early dry season daily evaporation was assessed according to the stage-2 theory of soil evaporation. The regression between cumulative daily $E$ and $\sqrt{t_d}$ is linear for all methods (Fig. 4). The derived $D_e$ values are the highest for the B16 method, ranging from 2.23 to 4.20 mm d$^{-1/2}$. The largest difference between the methods occurred in the late wet season in 2015 with $D_e$ values 2.91, 1.92, and 1.08 for the B16, Z16, and N18 methods (Fig. 4f). The N18 method has the lowest $D_e$ values except in mid dry season in 2015 (Fig. 4g). The estimated $D_e$ values were linearly related to the first day air temperature for the B16 and Z16 methods (Fig. S7). Overall, the $D_e$ values from the B16 method match most closely the reported range of $D_e$ from other studies of sandy soils (Brutsaert and Chen, 1995; Hu and Lei, 2021).

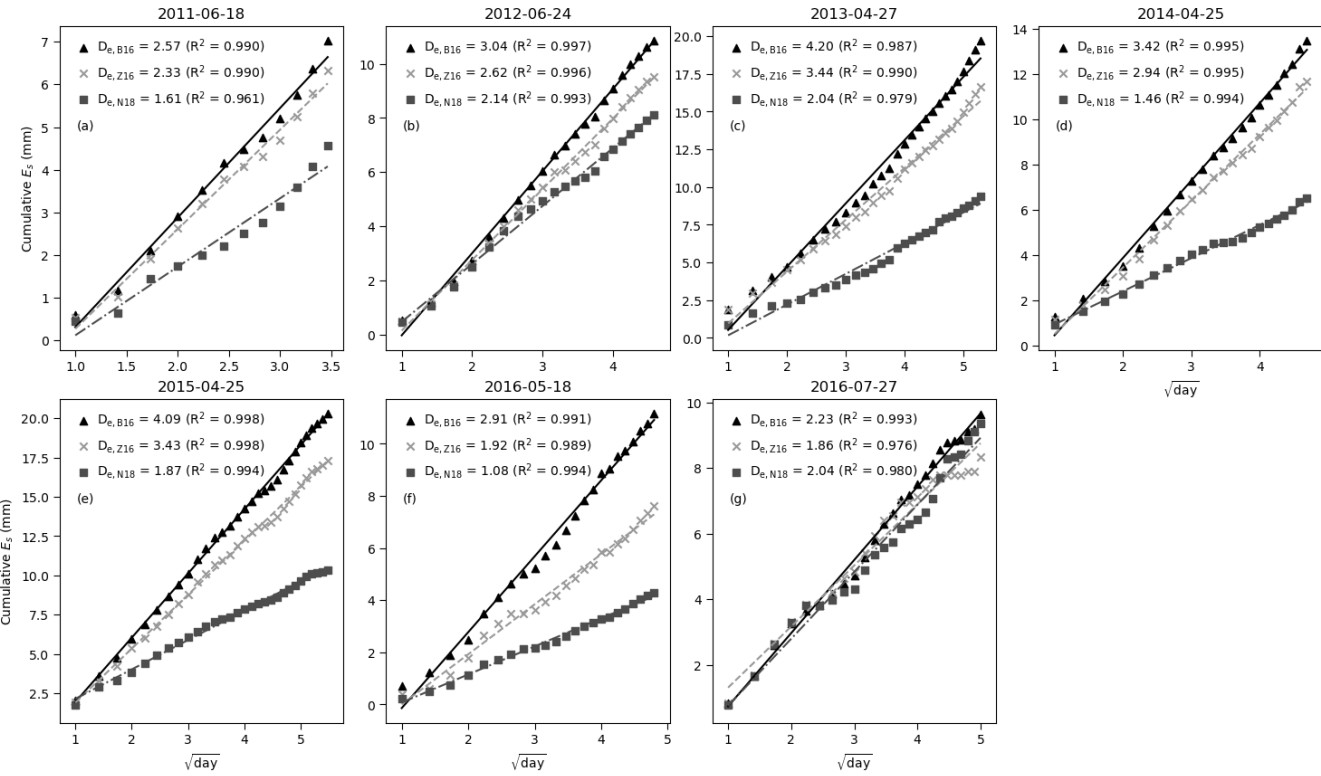

**Figure 4. Relationship between cumulative daily soil evaporation and (day)$^{1/2}$ for the stage-2 evaporation events for B16, Z16 and N18 methods. The slope of the regression line is soil desorptivity. Titles indicate the start date for each dry-down event.**

## 3.2 Monthly tree and grass transpiration

A comparison of the three different monthly $T$/ET estimates shows that $T$/ET according to the N18 method is consistently higher than the $T$/ET from the other two methods during the wet seasons (Fig. 5a). The B16 and Z16 methods have similar $T$/ET seasonality during the wet seasons, whereas the N18 $T$/ET is higher than the other methods during the late wet seasons. The monthly $T$/ET from N18 is nearly constant during the wet season in 2011, whereas the B16 and Z16 $T$/ET show decreasing trend after the early season peak. The largest difference between $T$/ET estimated with the Z16 and B16 methods occurred from March to June 2015 (Fig. 5a). During that period, the monthly GPP decreased, while $T$/ET increased according to all methods. The B16 soil desorption was closest to published values during this period (Fig 4f), suggesting that Z16 and N18 overestimated $T$/ET during the late wet season of 2015.

Estimates of monthly $T$ were similar based on the B16 and Z16 methods, but consistently higher based on the N18 method (Fig 5b). The rainy season began 48 days later in 2014 than in 2012 (Table S1). This delay is consequently reflected in the

monthly course of $T$ (B16 method) in 2014, which lagged behind that of 2012 until January (Fig. 5b). The dry spell in 2011 is clearly shown by reduced $T$ and ET during this period. The soil moisture-based estimate of grass transpiration shows a similar trend to estimated $T$ in 2012 and 2013, whereas the $T_g$ trend is less variable than $T$ trend in 2014 and 2015 (Fig. 5b). In 2014 the $T_g$ was nearly constant during the wet season despite increasing grass NDVI (Fig. 5b-c).

Both the tree and grass peak NDVI values are lower during the drought year compared to the wet years 2013 and 2014 (Fig. 5c). The grass dieback is reflected in low grass NDVI after the initial increase during the early drought year. In addition, the $T_g$ is decreasing during this period. The second peak in grass NDVI is similar in magnitude to the first suggesting modest grass growth after the dieback. The grass NDVI and $T_g$ decrease during the mid and late wet season, while the tree NDVI and $T$ stay

nearly constant. This means that the tree contribution to the total transpiration increases from mid-wet season onwards in 2015.

The monthly $T$/ET to LAI relation was scattered for all methods (Fig 6a). The $T$/ET was higher for the Z16 method than B16 at the low LAI values. The VPD response of monthly GPP/$T$ was most non-linear for the B16 method and least non-linear for the N18 method (Fig. 6b). The monthly LAI to $T$ relation was linear for all methods and similar between the B16 and Z16

methods (Fig. 6c). The $T_g$ was increasing for higher LAI values in 2012 and 2013, whereas it was nearly constant at 15 mm month$^{-1}$ for a wide range of LAI values in 2014 and 2015 (Fig. 6d).

The monthly GPP and $T$ were linearly related (Fig. S8 $R^2 = 0.97$, $p < 0.001$), allowing for an estimate of an effective (constant) ecosystem water-use efficiency using a zero-intercept regression. The constant water-use efficiency was 2.83, 2.78, and 2.29

g C/kg H$_2$O for B16, Z16, and N18 methods (Fig. S8).

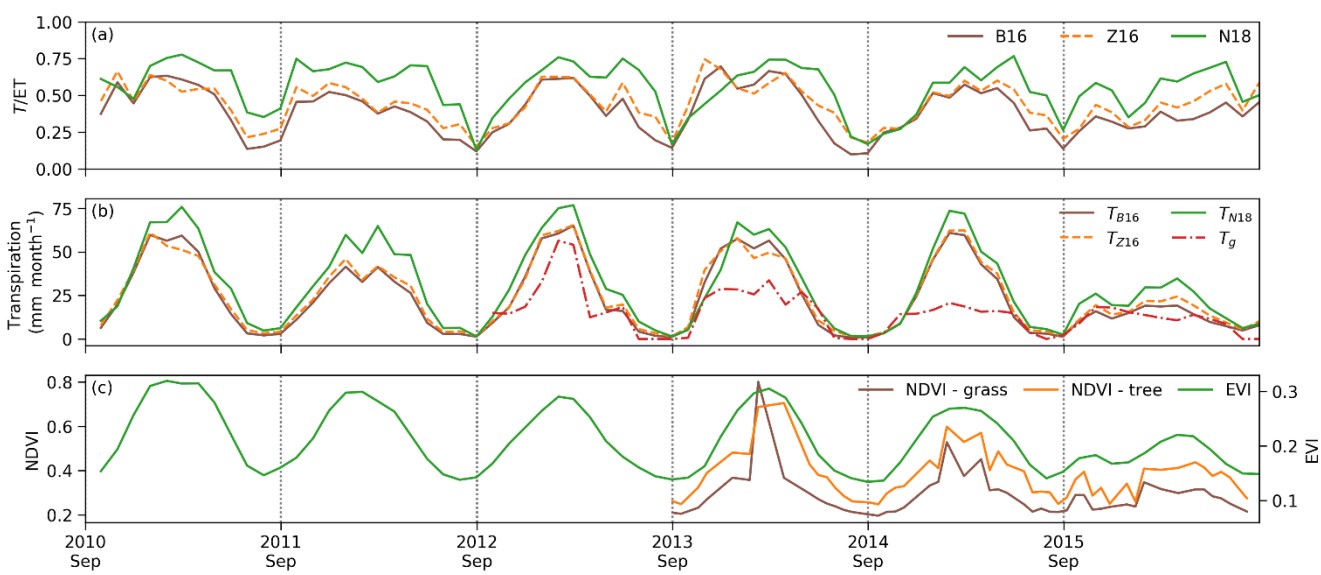

**Figure 5. (a) Time series of monthly *T*/ET estimated with the B16, Z16, and N18 methods. (b) Time series of monthly *T* and $T_g$. (c) Time series of 16-day MODIS EVI and Landsat 8 NDVI for grasses and trees. The dotted vertical line indicates the start of the hydrological year (September 1st).**

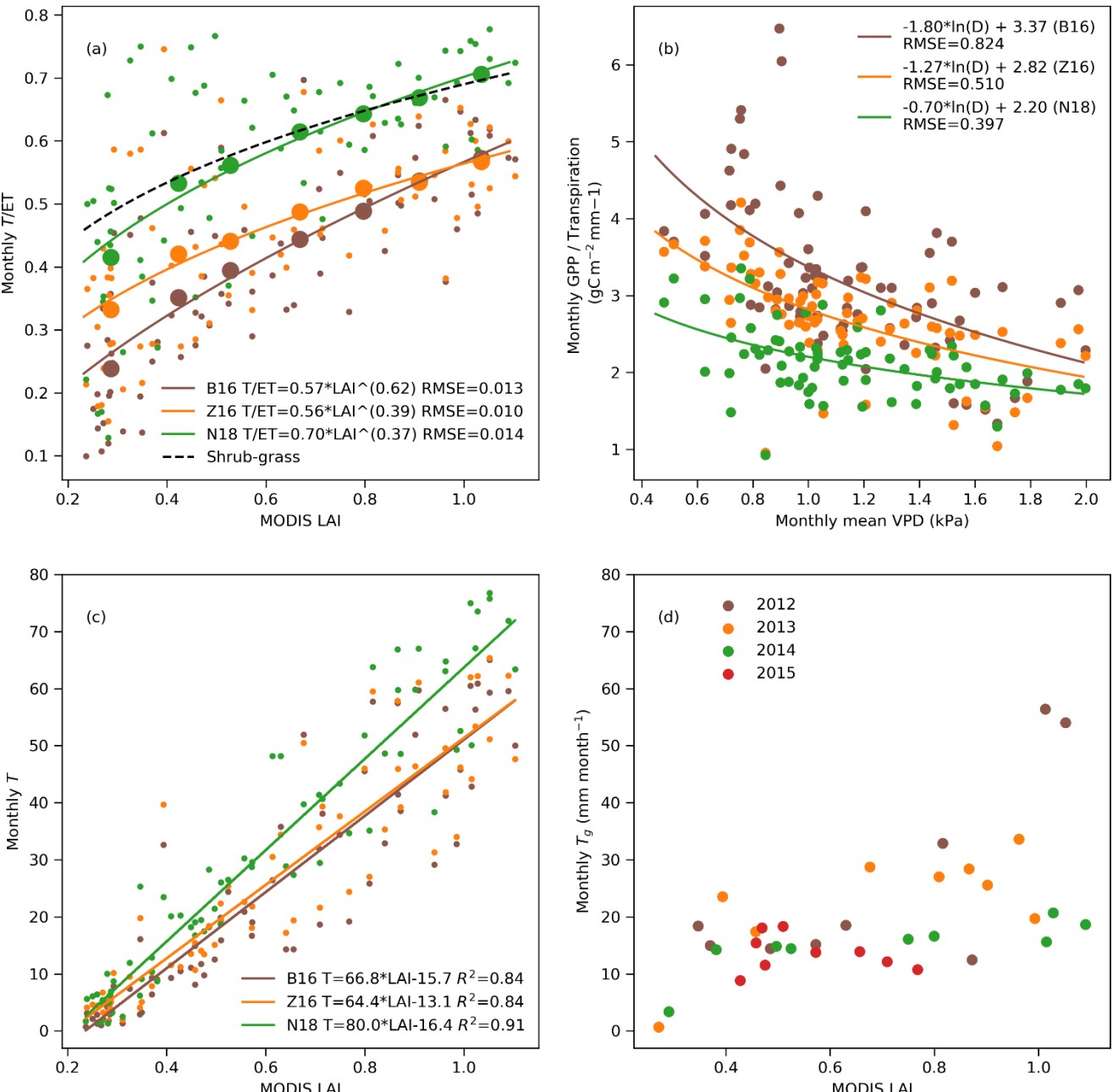

**Figure 6. Relationship between monthly *T*/ET and MODIS LAI. The black line is the relationship for shrub and grass ecosystems that is based on measured LAI values (Wei et al., 2017). The equal-width bins are fitted using the equation a\*(LAI)^b. (b) The relationship between monthly GPP/Transpiration and monthly mean VPD. The fit line is a\*ln(VPD)+b. (c) Relationship between monthly *T* and MODIS LAI. (d) Relationship between monthly *T_g* and MODIS LAI where colors indicate data from the four different years.**

### 3.3 Interannual variation

**Table 2. Annual sum of water balance components for each hydrological year (September to August). The total uncertainty (Eq. 4) is indicated for ET after the $\pm$ sign. $ET_N$ is the annual nighttime evapotranspiration. Transpiration and evaporation estimated using the B16 method are shown. The tree transpiration ($T_{tree}$) was calculated by subtracting $T_g$ from $T$. The PET was determined from Eq. 1. The EBC slope stands for the slope of the energy balance closure with ordinate, defined by measured $R_n\text{-}G$ and abscissa defined by the sum of the measured latent and sensible heat fluxes.**

| Year | $P$ | ET | $ET_N$ | $P$-ET | $T$ | $T_g$ | $T_{tree}$ | $E$ | $E_s$ | $I_t$ | $\Delta\Theta_{1m}$ | $T$/ET | PET | $EVI_{max}$ | EBC-slope |
|---|---|---|---|---|---|---|---|---|---|---|---|---|---|---|---|
| | (mm yr$^{-1}$) | (mm yr$^{-1}$) | (mm yr$^{-1}$) | (mm yr$^{-1}$) | (mm yr$^{-1}$) | (mm yr$^{-1}$) | (mm yr$^{-1}$) | (mm yr$^{-1}$) | (mm yr$^{-1}$) | (mm yr$^{-1}$) | (mm yr$^{-1}$) | | (mm yr$^{-1}$) | | |
| 2010–2011 | 628 | 658 $\pm$ 8 | 68 | −30 | 341 | - | - | 317 | 230 | 87 | - | 0.52 | 1133 | 0.32 | 0.75 |
| 2011–2012 | 577 | 608 $\pm$ 10 | 68 | −31 | 255 | - | - | 353 | 279 | 74 | - | 0.42 | 1123 | 0.30 | 0.80 |
| 2012–2013 | 672 | 667 $\pm$ 8 | 78 | 5 | 325 | 237 | 88 | 342 | 260 | 82 | 14 | 0.49 | 1109 | 0.29 | 0.85 |
| 2013–2014 | 580 | 600 $\pm$ 6 | 81 | −20 | 339 | 205 | 134 | 261 | 171 | 90 | 12 | 0.56 | 1039 | 0.31 | 0.81 |
| 2014–2015 | 636 | 642 $\pm$ 11 | 85 | −6 | 300 | 140 | 160 | 341 | 244 | 97 | 3 | 0.47 | 1057 | 0.27 | 0.83 |
| 2015–2016 | 508 | 463 $\pm$ 6 | 58 | 45 | 154 | 131 | 23 | 309 | 240 | 69 | −1 | 0.33 | 1038 | 0.22 | 0.84 |
| Mean | 600 | 606 | 73 | -6 | 286 | 178 | 101 | 320 | 237 | 83 | 7 | 0.46 | 1083 | 0.28 | 0.81 |
| SD | 53 | 69 | 10 | 26 | 66 | 44 | 52 | 31 | 34 | 9 | 6 | 0.07 | 40 | 0.03 | 0.03 |

Annual $P$ was close to or above the mean annual rainfall of 591 mm yr$^{-1}$ (adjusted here for the undercatch) for every year except in 2015, which was an extreme drought year in South Africa (Table 2). The EC-measured annual ET was close to annual $P$ for all years (Table 2). The annual $P$–ET ranged from −31 to 45 mm yr$^{-1}$, and it was inversely related to the annual maximum EVI ($R^2 = 0.87$, $p = 0.007$). The annual change in soil water storage was small (1 to 14 mm yr$^{-1}$) compared to the variation in

other water balance components and unrelated to the annual $P$–ET ($R^2 = 0.45$, $p = 0.332$). The frequent evening and nighttime precipitation resulted in nighttime evapotranspiration ($ET_N$), which varied from 58 to 85 mm yr$^{-1}$ (12% of annual ET). The annual $P$–ET would be positive for all years if $ET_N$ were assumed to be zero. The annual estimated rainfall interception ranged from 69 to 97 mm yr$^{-1}$, linearly related to $ET_N$ ($R^2 = 0.75$, $p = 0.025$).

The estimated annual $T$/ET ratio varied from 0.33 to 0.56 for the B16 method (Table 2). The annual $T$/ET from Z16 was similar to the B16 method with a 0.04 higher six-year mean, whereas the N18 $T$/ET mean was 0.15 higher than the B16 mean (Table S2). The $T$/ET in 2011 was similar to other wet years for the N18 method but reduced compared to other wet years for the B16 and Z16 methods (Table 2 and S2). The annual $T$/ET and $T$ were linearly related to early wet-season storm frequency for the B16 and Z16 methods (Fig. 7), while the relation of early season $P$, mid wet season $P$, annual $P$ or mean annual EVI with annual $T$/ET, and $T$ were more scattered (Fig. S9, S10, and S11). For the N18 method, the annual $T$ was linearly related to mid wet season $P$ (Fig. S10), and annual $T$ and $T$/ET were linearly related to mean EVI (Fig. S11).

Annual transpiration was nearly constant at $326 \pm 19$ mm yr$^{-1}$ for the four years with frequent early-season rainfall (Table 2). The annual $T$ was similar for wet years 2012 and 2014, despite 97 mm lower $T_g$ and the late start of rainy season in 2014. The lower $T$ in 2011 is explained by the infrequent early-season rainfall and the two-week dry spell in January. The average dry-season transpiration was 9 mm (over three months)$^{-1}$ for the B16 method (Table S3), which suggests a minimum tree transpiration of 36 mm yr$^{-1}$. The annual $T$–$T_g$ representing tree transpiration ranged from 88 to 160 mm yr$^{-1}$ for the three wet years (Table 2). The annual $T_g$ was similar for 2014 and 2015 despite significantly lower grass NDVI in 2015 (Fig. 5c). Therefore, during the drought year, $T_g$ might be an overestimate due to contribution from soil evaporation, which would explain the low value (23 mm yr$^{-1}$) of tree transpiration.

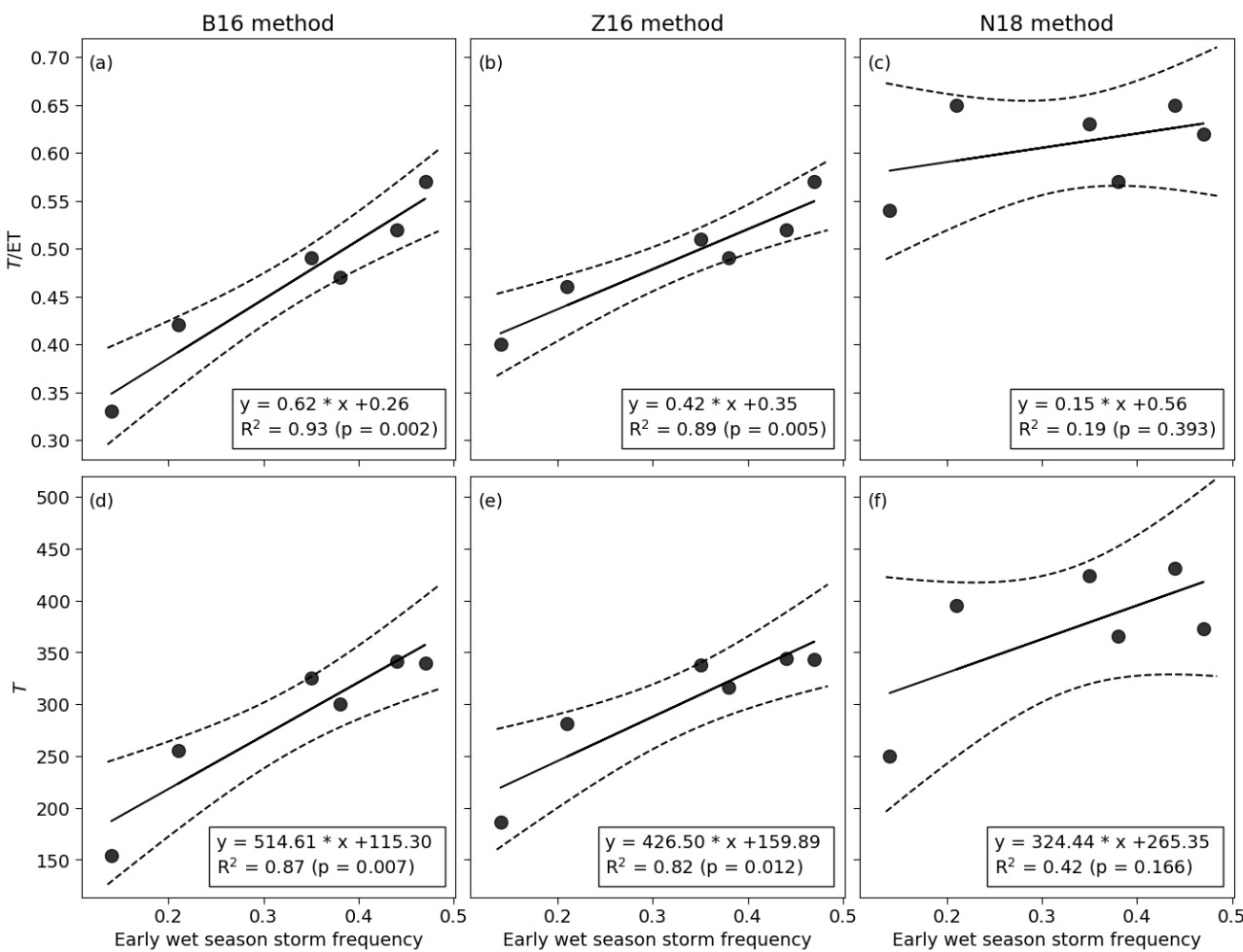

**Figure 7. Relationship between early wet-season storm frequency and annual *T*/ET and *T* for the B16, Z16 and N18 methods. Dashed lines demarcate the 95% confidence intervals.**

## 4    Discussion

At annual timescales, $P$ was approximately equal to annual ET, consistent with other studies from sites with similar annual rainfall (Gwate et al., 2018; Scholes and Walker, 1993). Annual $T$ was nearly constant ($326 \pm 19$ mm yr$^{-1}$) during the four years with frequent early wet-season precipitation (Table 2), as has been found in different types of forest ecosystems (Oishi et al., 2010; Tor-ngern et al., 2017; Ward et al., 2018). However, it was lower in years with infrequent early wet-season rainfall producing intermittent dry spells. The variation in annual $T$ and $T$/ET was explained by the variation in early wet-season storm frequency (Fig. 7). The monthly $T$ was linearly related to LAI, whereas the $T$/ET had an expected non-linear relation to LAI (Fig. 6a,c). The constancy of annual $T$ reflects moderate water stress of $C_4$ grasses shown by similar $T_g$ during the two wet years (Table 2). The annual $T$ was not reduced during the year with late rainy season start and lower $T_g$, showing that the trees can use the water that is not transpired by the grass layer (Fig. 5b). However, during the extreme drought year the peak grass and tree NDVI was lower than during the wet years (Fig. 5c). The $T_g$ was decreasing and grass regrowth was modest after the grass dieback from mid to late wet season, while the tree NDVI was constant, showing greater contribution of tree transpiration during the late wet season (Fig. 5b,c). The rainfall timing control on annual $T$ and the grass dieback and regrowth dynamics show that the grasses' early season development foreshadows the interannual variation in $T$ and $T$/ET, which reflects the fact that the temporal distribution of $P$ is a good predictor of the productivity at these highveld grasslands (Swemmer et al., 2007).

### 4.1    Transpiration

The small interannual variation in transpiration is likely due to moderate water stress of $C_4$ grass layer and tree ability to use water from shallow and deeper soil layers, depending on water availability. The photosynthesis reduction in $C_4$ grass is more related to non-stomatal limitations compared to $C_3$ grass, which is predominantly limited by stomatal control (Ripley et al., 2010). Under South African field conditions over one growing season, the $C_4$ grass layer tended to maintain a constant difference between predawn and midday leaf water potential, with similar transpiration at rain-fed and irrigated pot trials (Taylor et al., 2014). The annual $T_g$ at Welgegund was similar during the two wet years, but lower during the year with late start of the rainfall. During the 2015 drought year, the grass cover underwent a dieback-regrowth cycle, tree peak NDVI was less than during wet years, and the annual $T$ was reduced by 51% from the mean of wet years (Fig 5b and c). The tree contribution to the total transpiration increased from mid-wet season onwards during the drought year as $T_g$ decreased while $T$ increased (Fig. 5b). Similar grass growth pattern was also observed in Kruger National Park, where the grass biomass decreased and vast areas were barren during the drought, but the grasses quickly recovered once the rains returned (Wigley-Coetsee and Staver, 2020).

The trees at Welgegund likely rely more on topsoil water when available, switching to greater reliance on deeper layers during dry seasons and droughts. Similar tree water-use pattern has been observed using stable isotopes at South African savanna (Priyadarshini et al., 2016). Welgegund is located at the wet end of the distribution range of the dominant tree species *Vachellia*

*erioloba*. The estimated radiocarbon age of these trees is approximately 20 years (Steenkamp et al., 2008). The roots of *V. erioloba* are deep and reportedly extend to a depth of up to 60 m (Jennings, 1974); in one study, the roots absorbed 37% of the transpired water below a depth of 1 m (Beyer et al., 2018). In addition, the horizontal extent of the roots of this species can exceed 20 m (Wang et al., 2007). When soil moisture profile measurements at the site were installed 15 m away from the

nearest tree, tree roots were observed at a 0.4 m from the surface and deeper. The mean annual tree transpiration was 127 mm yr$^{-1}$ (40% of annual $T$) for the three wet years (Table 2). This estimate is similar to the tree transpiration of 126 mm yr$^{-1}$ at a site in South Africa (Nylsvley) (P = 586 mm yr$^{-1}$) with a 30% tree cover and shallower tree roots (measurements and modeling; Scholes and Walker, 1993). At a savanna site in this region (P = 241 mm yr$^{-1}$), *V. erioloba* (5 m tall) had an annual/dry-season transpiration ratio of 6.5 (3.9/0.6 mm d$^{-1}$, Tfwala et al., 2019). Multiplying this ratio by the 9 mm dry-season tree transpiration

in our study results in 59 mm yr$^{-1}$ annual tree transpiration. This shows that the dry season-based tree transpiration estimate is lower than the $T$–$T_g$ estimated that combines eddy-covariance estimate of $T$ and soil moisture-based $T_g$. The $T$–$T_g$ tree transpiration estimate includes any error made in the $T_g$ estimation. The overestimation of $T_g$ likely explains the low tree transpiration during the drought year. The constant soil evaporation removed from $T_g$ estimate may not be adequate during the drought year leading to soil evaporation contribution to $T_g$. The $T_g$ estimation algorithm also has lower performance in dryer

soils (Jackisch et al., 2020). The estimated long-term water-use efficiency was 2.83 and 2.78 g C/kg $H_2O$ for B16 and Z16 methods. This is somewhat higher than the wet season value of 2.4 g C/kg $H_2O$ (Z16 method) at a C$_4$ grassland site in southeastern Arizona, USA (Scott et al., 2021), and the field-scale long-term grass community value of 2.15 g C/kg $H_2O$ for the aforementioned shallower rooted trees and 30% tree cover savanna at Nylsvley (Scholes and Walker, 1993).

During water-stressed years, the partitioning of tree and grass contribution to LAI and $T$/ET may be needed to derive meaningful relations at the monthly scale. The drought year was characterized by the different grass and tree NDVI trends (Fig. 5c) and the nearly constant grass transpiration for a wide range of LAI values (Fig. 6d). New remote sensing products may be able to separate these contributions, as shown by a recent study that successfully separated tree and grass leaf area using a canopy height model, Sentinel vegetation indexes (10 m spatial resolution), and a Sentinel radar band during the 2015

drought in Kruger National Park (Urban et al., 2018). The effect of dieback-regrowth on annual transpiration is also interesting, as a stochastic model based on measured precipitation statistics with explicit bare soil, grass, and tree cover showed that vegetation dynamics had little effect on annual transpiration (Williams and Albertson, 2005).

Water availability for grass is the dominant factor in transpiration at Welgegund. The annual $T$/ET range (0.33 to 0.56) in

Welgegund is slightly wider than the mean annual $T$/ET range of 0.35 to 0.46 at a C$_4$ grassland site in southeastern Arizona, USA (P = 317 mm yr$^{-1}$), where the annual $T$/ET correlates with annual $P$ and mean LAI (Scott et al., 2021). At Welgegund, the early season rainfall frequency was a better predictor of annual $T$ than annual $P$ or mean EVI (Fig 7, S10, and S11). These relations might be due to the heavy grazing at the site, which limits peak EVI and emphasizes the early season grass development. The analysis of the different C$_4$ grass species during the 2014–2016 South African drought suggests that their

bundle sheath morphology explains the differences in drought tolerance (Wigley-Coetsee and Staver, 2020). Therefore, it is difficult to generalize whether the invariance of annual transpiration during the wet years would hold for sites with higher grass LAI or different grass species composition.

## 4.2 Uncertainty

The six-year mean ET/$P$ ratio was 1.0 at Welgegund (Table 2), slightly higher than a long-term ET/$P$ ratio of 0.96 at a C$_4$ grassland in Arizona, USA (Scott and Biederman, 2019). The lowest annual $P$–ET was –31 mm yr$^{-1}$, which is more negative than the estimated annual ET uncertainty but less than the uncertainty related to ET$_N$ gap-filling (Table 2). Due to frequent afternoon and nighttime precipitation, the ET$_N$ was 12% of the annual ET. The ET$_N$ values here may appear high but are commensurate with reported values for forested ecosystems (Novick et al., 2009) in regions with higher precipitation and LAI.

The gap-filled ET$_N$ may be an overestimate because only 30% of the values were measured and these values were determined during high wind speeds ($u_* > 0.28$ m s$^{-1}$).

The ratio of annual ET uncertainty to annual ET was 1.3%, which is lower than the 5 to 9% range reported from eddy covariance ET measurements from a cultivated area in Benin (Mamadou et al., 2016). The difference can be ascribed to

different error terms used in the uncertainty estimation. The mid-dry season ET ranged from 45 to 68 mm (3 months)$^{-1}$ (mid-dry monthly value multiplied by three) at this cultivated site in Benin that has isolated trees (height < 10 m) and bare soil during the dry season (Mamadou et al., 2014, 2016). This is higher than the 29 to 52 mm (three month)$^{-1}$ range measured in our study. These differences may be attributed to the relatively shallow water table (a depth of 3 m during the dry season) and the higher annual precipitation ($P$=1200 mm yr$^{-1}$) at the Beninese site.

## 4.3 ET partitioning methods

The B16 and Z16 transpiration estimates were more similar and closer to reported grassland $T$/ET values than the N18 estimate (Fig. 5a). The N18 estimate was also higher than the Z16 estimate at the C$_4$ grassland site in southeastern Arizona, USA (Scott et al., 2021). The annual maximum of monthly $T$/ET ranged from 0.57 to 0.67 for the four wet years (B16 and Z16 methods),

which is similar to the maximum value of 0.60 at a C$_4$ grassland site in southeastern Arizona, USA estimated using the ET partitioning method that does not assume equality between $T$ and ET (Scott and Biederman, 2017). The assessment of soil evaporation according to stage-2 theory (Fig. 4) showed that the B16 estimated $D_e$ matched most closely the reported range of $D_e$ (3 to 6 mm d$^{-1/2}$ ) from other studies of sandy soils (Brutsaert and Chen, 1995; Hu and Lei, 2021). The largest difference in $D_e$ between the B16 and Z16 methods occurred in the late wet season in 2015, when the Z16 $T$/ET was deemed an

overestimate based on the low $D_e$ value of 1.92 mm d$^{-1/2}$. The estimated $D_e$ values were linearly related to the first-day air temperature for the B16 and Z16 methods (Fig. S7). Similar dependence has been observed in laboratory conditions with full

wetting of sandy soil columns (Ben Neriah et al., 2014). For the Z16 method, a one-to-one $T$=ET line is fitted using quantile regression for each year combined with the intercept forced through zero, whereas for the B16 method the $T$=ET line is fitted using linear regression over the GPP $\times$ VPD$^{0.5}$ bins. Regarding the N18 method, $T$ was likely an overestimate for several reasons. The estimated $D_e$ values were much lower than the literature values, and the annual $T$/ET values had much higher range (0.54 to 0.65) than other methods and studies (Scholes and Walker, 1993; Scott et al., 2021). In addition, the tree transpiration estimated using N18 by subtracting estimated grass transpiration from N18 estimated annual $T$ results in average annual tree transpiration of 193 mm over the three wet years, which is much higher than reported in the literature. The N18 algorithm does not use measured soil moisture in the training period, but instead uses $P$ and ET water balance, which may explain the small interannual variance of the maximum $T$/ET values (Nelson et al., 2018).

The low surface soil moisture values during $T$=ET periods and their concentration during the rainy season give assurance that the annual fitted $T$=ET lines correspond to periods when $T$ equals ET (Table S1). The annual $T$=ET line could be predicted using the rainy season length, except in 2011, which experienced the earliest green-up of trees and the highest number of $T$=ET moments outside the rainy season (Fig. 4a, Table S1). The water balance analysis, focused on monthly and annual timescales using the ET partition methods, has shown good agreement with independent estimates (Berkelhammer et al., 2016; Zhou et al., 2018). Berkelhammer et al. (2016) showed that a 3-day running mean of the half-hour $T$/ET estimates reduced the root-mean-square difference between the Berkelhammer method and the isotopic estimate of $T$/ET to $\leq 0.2$. Therefore, the random error of the monthly means of the half-hour $T$/ET estimates in this study can be assumed small. For a Mediterranean tree-grass savanna, $T$/ET was shown to rarely exceed 0.8 (Perez-Priego et al., 2018). In contrast to the Mediterranean site, the Welgegund site has sandy soil, deep-rooted trees, and no clay horizon close to the soil surface. More importantly, the mean surface soil moisture was 0.1 m$^3$m$^{-3}$ or below for the half-hour runs when $T$=ET at Welgegund. This low soil moisture resulted in small diffusion-limited soil evaporation and thus periods when $T$ equals ET. This is another independent confirmation of the partitioning of ET into $E$ and $T$ (even at such short timescales).

## 5    Conclusion

The measurements reported here show that the annual transpiration is nearly constant during years with frequent early-season rainfall but can be lower because the C$_4$ grass cover reacts to dry spells. Trees at the site are likely able to use water that is not used by the grass layer and use water from deeper layers during extreme drought. Our work highlights precipitation control over $T$ and the annual variation in the $T$ to LAI relationship. These results can be used to assess the water resources and fodder production of grassland grazing systems. Although further work is required to determine the generality of these conclusions to other savanna systems, the methodologies developed and tested here can be employed when investigating a wide range of arid and semi-arid ecosystems experiencing water shortages in times of drought.

**Data availability**

The data used in this study are available online https://doi.org/10.6084/m9.figshare.11322464 (Räsänen et al., 2019).

**Author contributions**

MR, RO, and GK designed the analysis; VV, MA, and JT performed the EC data processing. MA, VV, PB, JT, PVZ, MJ, SS,
TL, LL, MK, JR conducted the measurements. All authors contributed to the final version of the manuscript.

**Competing interests**

The authors declare that they have no conflict of interest.

**Acknowledgments**

This work was supported by the Finnish Meteorological Institute, North-West University, the University of Helsinki, the
Finnish Academy project Developing the atmospheric measurement capacity in Southern Africa, and the Finnish Centre of
Excellence, grant no. 272041. This publication forms part of the output of the Biogeochemistry Research Infrastructure
Platform (BIOGRIP) of the Department of Science and Innovation of South Africa. This work was partially funded by the
European Commission through the project "Supporting EU-African Cooperation on Research Infrastructures for Food Security
and Greenhouse Gas Observations" (SEACRIFOG; project ID 730995). Financial support for R. Oren was provided by the
Erkko Visiting Professor Programme of the Jane and Aatos Erkko 375th Anniversary Fund, through the University of Helsinki.
G. Katul acknowledges partial support from the U.S. National Science Foundation (grant numbers NSF-AGS-1644382, NSF-
AGS-2028633 and NSF-IOS-175489). The corresponding author wishes to thank Janne Heiskanen for discussions on the
Landsat data analysis. The authors thank the ranchers for their help in the setup and instrument maintenance.

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
