# Peer review of "The effect of rainfall amount and timing on annual transpiration in a grazed savanna grassland"

_Hydrology and Earth System Sciences, 2021_

## Author Comment (AC1)

**Response to Russell Scott (Referee #1)**

In this paper, the authors examine the transpiration of a grazed savanna in Africa to determine what are the controls on annual ET partitioning. They do this by using 6 yrs of eddy covariance data and three different partitioning techniques. They conclude that early season rainfall timing strongly controlled annual T/ET by affecting the growing season dynamics primarily of grasses rather than the trees.

I found this study very interesting and generally, sound. I think it will be potentially of great interest to the readers of this journal. However, I found the presentation of the results confusing at times and would recommend a thorough restructuring of them. Many of the authors' conclusions are conjectures about the grass functioning with little to back them (i.e., you've got T and ET but not Tgrass and Ttree).

We thank Dr. Russel Scott for all the helpful comments and suggestions. Thanks to your suggestions we have considerably revised the manuscript results and presentation. As detailed below, the main changes we propose are:

i) including estimates of daily grass transpiration based on the soil moisture profile measurements available for the last four years using the RWU method (Jackisch et al., 2020).
ii) evaluating the dry season ET partitioning results from all the partitioning methods based on the stage-2 soil evaporation theory (Hu and Lei, 2021)
iii) restructuring the results to the following sections: Site meteorology and ET partitioning, monthly T and T/ET, and interannual variation.

Here are a few suggestions and comments that hopefully may guide a restructuring of the paper:

1. The paper needs a deeper look into the controls on the total T and T/ET. Ultimately, this has got to be about water availability, right? Rain event frequency is really an indirect way of looking at it. It says nothing about the total amount of water and where it is located (shallow or deep). Certainly, storm depth must be a critical factor in how frequency is translated into water availability. Since you've got the data to do it (E, T, GPP, LAI, soil moisture) can you better unpack the seasonal pattern, showing in greater detail how summed T and E and soil moisture evolve through the early to middle part of a growing season contrasting a normal year with a dry one? You could look at the monthly level data, but you should be able to do this on a daily scale for the TEA or Berkelhammer results if you wanted to show the finer dynamics. Also, what about the E dynamics? Does storm frequency have an influence in the amount of E?

Thank you for this suggestion. We have estimated the daily grass transpiration based on the soil moisture profile measurements available for the last four years (Jackisch et al., 2020). For the wet years 2012 and 2013, the deep soil layer (60cm) gets wet and the estimated grass transpiration is over 200 mm/year. In 2014, the deeper soil layer stays dry and grass transpiration is reduced while total transpiration is similar to the wet years. In the drought year, the grass transpiration is decreasing while the total transpiration is increasing during the middle of the wet season, suggesting higher relative contribution of tree transpiration. We will add a new figure showing the soil evaporation dynamics by

presenting the stage-2 soil evaporation analysis of the last precipitation event of each year for all methods used. The cumulative daily soil evaporation is regressed against day^(0.5). Linear scaling is expected and the derived soil desorption values can be compared to published values. The soil desorption values, representing the soil evaporation after rainfall events, are linearly correlated with the initial air temperature for the Berkelhammer and uWUE method. This relation has also been observed in laboratory conditions (Ben Neriah at al., 2014).

2. There are lots of inferences about grass and tree functioning, but little data about this is shown in the results. Is there a way you can use the remote sensing and the monthly data to make your case more strong? E.g., you write that there aren't many LAI changes for the trees so LAI is really indicative of the grass LAI. You're also saying that the C4 grasses control their water use by dying back or growing new leaves. If so, is there a way to use T/ET (or maybe better, just T) and LAI to show this more clearly?

We will present the monthly time series of $T_{grass}$ and T for all methods. Also, we will show in the fig. 6 the monthly T/ET, T and $T_{grass}$ as a function of MODIS LAI. The $T_{grass}$ scales linearly with increasing LAI during the wet years 2012 and 2013. However, during the drought year, monthly $T_{grass}$ is limited to 15 mm/month for a range of LAI values, showing that the T is increasing together with an increase in tree LAI. The new results show that tree transpiration can vary among years and variation in wet season LAI may reflect those of the both grass and tree fractions. We will update the manuscript according to these new results.

3. The introduction is underdeveloped. What is missing are the previous results that lead to expectations of what you might find here. There are quite a few studies cited for semi-arid systems but what have you learned from them that help guide this analysis?

Thank you for this comment. We will update this section with field results from similar sites to better explain what we expect to find. At the least, we will include (1) the study by Swemmer et al. (2007) showing how number of precipitation events and their distribution are better predictors of above-ground productivity than total rainfall, and (2) the remote sensing analysis showing a positive relationship with grass cover and rainfall frequency (D'Onofrio et al., 2019).

Text specific suggestions and questions:

Title: Is it a savanna or grassland or both?

It's both. A grassland with typical savanna grass and tree species.

P1.

L 27-29. Is there data to support these claims about grass and tree functioning?

We will modify these statements according to the new analysis of the results.

L30. What is an anomalous monthly T/ET relation?

This referred to the Fig.6d pattern. We will remove this sentence.

L31. How can drought be reasonably described by P timing alone? Storm depth has got to play a role as it ultimately is about water availability and its timing.

We will revise this sentence. The intension was not to say that the drought is described by the P timing alone. The drought year is characterized by 11 months of infrequent rainfall. The interesting result is that the annual T and T/ET based on both Berkelhammer and uWUE method estimated T are best explained by the early season rainfall.

P.2

L12. You might also consider this paper which also talks about monthly T/ET dynamics in semiarid systems: Scott, R. L., & Biederman, J. A. (2017). *Geophysical Research Letters*,*44*(13), 6833-6840.

Thank you. We will included it in the revision.

L23. Maybe say "partially decoupling" as 37% isn't huge.

Corrected.

P.3

L5. These papers that compared approaches could be cited here: Berkelhamer et al. 2016, Scott et al. 2021, Nelson et al. 2020.*Global change biology*, *26*(12), 6916-6930.

We will include these references.

L11-L13. I don't understand how using both LAI and EVI allowed you to quantify the dynamics of the grasses and trees.

We will change this sentence. The early season EVI can be used estimate the tree green-up dates but during the wet season LAI and EVI are composite of grass and tree leaf area. The LAI, T and $T_{grass}$ enable better to quantify the dynamics of grasses and trees.

L19. "farm" or "ranch"?

Changed to ranch.

P.5

L20. "verified" or "computed"? Verified with what?

Computed

P.6

L10. averaged over what time period?

Daytime data for the whole six-year period.

P.9

L11. You need to sum 1/2 hr T's and ET separately and then take their ratio. You can't just take the average T/ET.

Thank you for this correction. We have now calculated the daily mean T/ET using each method and then used this daily T/ET according to your suggestion. For example, monthly T is sum of daily values of T. We do this on daily scale because the half-hour T/ET series from Berkelhammer and uWUE methods are discontinuous.

L25. I'm wondering why this was done across all years. I would think that the same reasons that you fit the Berkelhamer approach yearly should apply equally here. Yearly changes in ecosystem structure/lai should warrant a yearly fit.

We were following the uWUE method as it has been reported in the previous publications. However, to facilitate better comparison to the Berkelhammer method, we have now calculated the uWUE method also using the yearly fit.

L27. "…of each month." This is confusing. I thought uWUEp was computed for the 6 yrs and uWUEa was computed monthly?

This is a mistake. We will revise this. We have now computed uWUEp for each year.

P10.

Table 1. As you've already described the methods in the text. This table is superfluous. Suggest omitting.

This Table was suggested by the reviewer of the earlier version of this manuscript. We prefer to keep it for a general reader.

https://hess.copernicus.org/preprints/hess-2019-651/

L14-16. Was there a reason for using both EVI and LAI? I've always found the essentially the same information content in each signal. For simplicity in the presentation of the results, you might consider using only LAI.

The signals are slightly different and the green-up detection is sensitive to those changes. Because the EVI is the preferred product to be used for green-up detection, we kept both products (Adole et al, 2018).

Table 2. For the T and E columns. What method is this or is this an average between the three? Also, see Fig. 5b…no method given.

It's the Berkelhammer method. We will specify each T in the revised manuscript. We will show $T_{grass}$ and T from each method in the Fig. 5.

P16.

L7-9. This is a possibility but isn't it also possible that T/ET in the late rainy season goes up as the soil dries and E becomes negligible?

We have compared the estimated E from each model during this period using the stage-2 soil evaporation theory. After a precipitation event at the end of May, the soil is dry for stage-2 conditions. The soil desorption is the slope of the cumulative daily soil evaporation and day^(0.5). This relation is expected to be linear. The soil desorption values are 2.91, 1.92 and 1.08 mm/day^(0.5) for the Berkelhammer, uWUE and TEA methods. The expected range for the soil desorption is from 3 to 6 mm /day^(0.5) for sandy soils (Brutsaert and Chen, 1995; Hu and Lei, 2021). This suggests that the Berkelhammer method produced the most plausible soil evaporation, but we will analyze this further by also looking at the LAI changes over this period.

L16. "due to a higher early-season precipitation frequency". Sorry to beleaguer the point, but the higher frequency may be a symptom rather than the cause of higher water availability.

We will omit the Fig.6c-d and focus the analysis on the $T_{grass}$ and T for all methods. The comparison was shown here because it was known that the early season P frequency is the best predictor of annual T and T/ET.

L29. It gets awkward to use the inverse. Why not present the usual WUE metric instead, making these numbers readily comparable to previous studies?

We will change these to the usual WUE units.

P18.

L2-3. As this section jumps back into the site water balance shown in 3.1, I found it confusing. You might change the organization of the results to one being about the water balance (talking P, ET, T, interception, Esoil, deltaS etc. and their variability) and the other being ET partitioning. Also, maybe adding a section that talks about the grass/tree dynamics separately to better support your claims.

This is a good suggestion. We will restructure the results by first showing the daily time series and ET partitioning results. Then the monthly T, T/ET and $T_{grass}$ for all methods, showing differences among method and distinguishing between tree and grass dynamics. The water balance and interannual variation is presented in the final section.

L11. See comment L16 above.

We failed to put this result to context in the discussion. The early season P frequency is the best predictor of annual T and T/ET. If we exclude the drought year, then it's the only significant predictor. It reflects the fact that temporal distribution of P affects the productivity (Swemmer et al. 2007). This result is interesting from the ecohydrological modeling perspective because at the Kalahari precipitation gradient (P = 300 to 950 mm/year) the mean storm depth is practically constant (9.5 to 10.3 mm/day) and only the precipitation frequency increases along the gradient (Porporato et al., 2003).

P19.

L10-13. In this summary of the results where is the evidence for this? I think this paper would really be improved if you could organize your results to better show this.

As stated above we have now estimated the grass transpiration and we are able to better show the grass and tree dynamics.

L22, Not clear what this sentence is here to address. On the surface, it says rainfall frequency is not important.

We will revise this part of the discussion.

L25. Where is this dieback - regrowth shown? Can you use EVI or LAI to show this?

It is shown in the Fig. 2e. Two local maximums in the EVI during the drought year. This kind of seasonality is not seen during any other year since 2001 for the EVI.

P21.

L5. I would delete this comparison. Using a BR from a higher annual PPT site isn't appropriate. Also, in order to use the BR to estimate ET you need to rely on H which may or may not be subject to commensurate errors.

We will delete this comparison.

L4-18. I'd suggest also considering, Scott, R. L., & Biederman, J. A. (2019). Water Resources Research, 55(1), 574-588 here. To me, the fact that ET ~= P is really solid evidence for the validity of your ET measurements so long as runoff (surface or deep) is negligible. Having an ET = P seems quite appropriate especially if you have those deep-rooted trees to capture any deeper infiltration.

Thanks. We will consider this comparison. Indeed, the decline in grass transpiration during the wet year 2014 seems to be compensated by the tree transpiration.

P22.

L 9-16. The Scott and Biederman 2017 paper using an entirely different method suggests a peak of T/ET ~= 0.60 -0.70 for a drier savanna site, similar to the results you have here.

We will include this reference.

L20. This is a discussion point, not a conclusion that comes from this paper.

We will change this statement.

References

Adole, T., Dash, J., and Atkinson, P. M.: Large-scale prerain vegetation green-up across Africa, 24, 4054–4068, https://doi.org/10.1111/gcb.14310, 2018.

Ben Neriah, A., Assouline, S., Shavit, U., and Weisbrod, N.: Impact of ambient conditions on evaporation from porous media, Water Resour. Res., 50, 6696–6712, https://doi.org/10.1002/2014WR015523, 2014.

Brutsaert, W. and Chen, D.: Desorption and the two Stages of Drying of Natural Tallgrass Prairie, 31, 1305–1313, https://doi.org/10.1029/95WR00323, 1995.

D'Onofrio, D., Sweeney, L., von Hardenberg, J., and Baudena, M.: Grass and tree cover responses to intra-seasonal rainfall variability vary along a rainfall gradient in African tropical grassy biomes, Sci Rep, 9, 2334, https://doi.org/10.1038/s41598-019-38933-9, 2019.

Jackisch, C., Knoblauch, S., Blume, T., Zehe, E., and Hassler, S. K.: Estimates of tree root water uptake from soil moisture profile dynamics, 17, 5787–5808, https://doi.org/10.5194/bg-17-5787-2020, 2020.

Hu, X. and Lei, H.: Evapotranspiration partitioning and its interannual variability over a winter wheat-summer maize rotation system in the North China Plain, Agricultural and Forest Meteorology, 310, 108635, https://doi.org/10.1016/j.agrformet.2021.108635, 2021.

Porporato, A., Laio, F., Ridolfi, L., Caylor, K. K., and Rodriguez-Iturbe, I.: Soil moisture and plant stress dynamics along the Kalahari precipitation gradient, 108, https://doi.org/10.1029/2002JD002448, 2003.

Swemmer, A. M., Knapp, A. K., and Snyman, H. A.: Intra-seasonal precipitation patterns and above-ground productivity in three perennial grasslands, 95, 780–788, https://doi.org/10.1111/j.1365-2745.2007.01237.x, 2007.

---

## Author Comment (AC2)

**Response to Referee #2**

 The manuscript "The effect of rainfall amount and timing on annual transpiration in a grazed savanna grassland" looks at the ecohydrolocal flux dynamics from a semiarid tree grass system. The study utilises a six year dataset of eddy covariance data which has obviously been well maintained, quality controlled, and is of high quality, as well as additional meteorological and remote sensing data. The study focuses on evapotranspiration (ET), as well as the partitioned plant transpiration, soil evaporation, and interception over the six year period, with one particularly dry year with significantly reduced ET and gross primary productivity (GPP).

 I found study particularly interesting in the scientific set up, however, the comparison of ET partitioning methods and the conclusions drawn from the T/ET dynamics seemed to dismiss the discrepancies between the methods and instead assume that one particular method was most accurate without given much substantial evidence as to why. Given the high uncertainty in partitioning methods (Nelson et al. 2020, Scott et al. 2020), it would be more rigorous to apply multiple methods and base the conclusions on patters which agree, or an independent evaluation as to why particular methods are likely to fail in certain situations. Given that the uWUE and Berkelhammer methods are methodologically very similar, a better analysis would be to use method with very different assumptions, such as one that avoids the T=ET assumption (e.g. Scott and Biederman 2017 or Perez-Priego et al. 2018).

We thank the referee for their supportive comments. We will add comparison of the dry season ET partitioning results from all the partitioning methods based on the stage-2 soil evaporation theory (Hu and Lei, 2021). We will also show the daily and monthly results for all methods and point out clearly the differences between methods.

We tested the use of Scott and Biederman 2017 method but the multi-year monthly correlations between ET and GPP were not statistically significant so we could not apply this method at this site. Perez-Priego et al. 2018 method depends on the constant C (eq. 9) that is only available for C3 plants. Applying this method at this site would be inappropriate because the $C_4$ pump efficiency modifying the photosynthesis model requires another parameter to be calibrated and used vis-à-vis the $C_3$ photosynthesis. For the same reason, the Perez-Priego et al. 2018 method was not applied at maize field in the method intercomparison study by Hu, X. and Lei, H. 2021.

 For example, one particular issue with the uWUE/Berlkelhammer methods is that the GPP*VPD^(1/2) to ET relationship is static throughout a year. In the case of a tree grass system, particularly when the grass is inactive for part of the year, the assumption that the ecosystem GPP*VPD^(1/2) to ET holds for the entire year may not be valid as the ecosystem fluxes shift from more tree to more grass dominated, which would then impact the inferred T/ET as the minGPP||ET|| could correspond to a period not consistent to the current state of the ecosystem. This is not to say the the uWUE or TEA estimates would be more correct either, but the low T/ET patterns seen in 2015 may also be underestimated by the Berkelhammer method. Indeed, the T/ET values from uWUE and Berkelhammer are significantly lower overall than the mean T/ET pattern from Wei et al. 2017 in Figure 6a and would be on the low end of what is reported from site level studies in Schlesinger and Jasechko 2014.

Testing the assumptions of the uWUE/Berkelhammer methods is beyond this study. However, we have compared the estimated E from each model during this period using the stage-2 soil evaporation theory. After a precipitation event at the end of wet season 2015, the soil is dry for stage-2 conditions. The soil desorption is the slope of the cumulative daily soil evaporation and $day^{0.5}$. This relation is expected to be linear. The soil desorption values are 2.91, 1.92 and 1.08 mm/$day^{0.5}$ for the Berkelhammer, uWUE and TEA method. The expected range for the soil desorption is from 3 to 6 mm /$day^{0.5}$ for sandy soils (Brutsaert and Chen, 1995; Hu and Lei, 2021). This suggests that the Berkelhammer method produced the most plausible soil evaporation, but we will analyze this further by also looking at the LAI changes over this period.

The annual T and T/ET from Berkelhammer and uWUE methods is best predicted by the early season P frequency. This fact, that temporal distribution of P affects the productivity, is supported by other fieldwork from a similar site (Swemmer et al. 2007).

While this paper does not set out to be an inter-comparison of ET partitioning methods from eddy covariance, I would recommend that at least all analyses utilise all three partitioning methods presented to determine if patterns are robust across methods, and possibly the addition of a fourth method which does not make the T=ET assumption. This would make the conclusions more robust and make the work much more useful to the wider community.

Thank you for this comment. We will present all three methods in the revised results and clearly show the similarities and differences. We have also estimated the grass transpiration from soil profile measurements (Jackisch et al., 2020) and used stage-2 soil evaporation theory to compare the dry season ET partitioning results from all the partitioning method (Hu and Lei, 2021).

References (included for information)

Nelson, J.A. et al. (2020) 'Ecosystem transpiration and evaporation: Insights from three water flux partitioning methods across FLUXNET sites', Global Change Biology. doi:10.1111/gcb.15314.

Perez-Priego, O. et al. (2018) 'Partitioning Eddy Covariance Water Flux Components Using Physiological and Micrometeorological Approaches', Journal of Geophysical Research: Biogeosciences. doi:10.1029/2018JG004637.

Schlesinger, W.H. and Jasechko, S. (2014) 'Transpiration in the global water cycle', Agricultural and Forest Meteorology, 189–190, pp. 115–117. doi:10.1016/j.agrformet.2014.01.011.

Scott, R.L. et al. (2020) 'Water Availability Impacts on Evapotranspiration Partitioning', Agricultural and Forest Meteorology, p. 108251. doi:10.1016/j.agrformet.2020.108251.

Scott, R.L. and Biederman, J.A. (2017) 'Partitioning evapotranspiration using long-term carbon dioxide and water vapor fluxes: New Approach to ET Partitioning', Geophysical Research Letters. doi:10.1002/2017GL074324.

References

Brutsaert, W. and Chen, D.: Desorption and the two Stages of Drying of Natural Tallgrass Prairie, 31, 1305–1313, https://doi.org/10.1029/95WR00323, 1995.

Hu, X. and Lei, H.: Evapotranspiration partitioning and its interannual variability over a winter wheat-summer maize rotation system in the North China Plain, Agricultural and Forest Meteorology, 310, 108635, https://doi.org/10.1016/j.agrformet.2021.108635, 2021.

Swemmer, A. M., Knapp, A. K., and Snyman, H. A.: Intra-seasonal precipitation patterns and above-ground productivity in three perennial grasslands, 95, 780–788, https://doi.org/10.1111/j.1365-2745.2007.01237.x, 2007.

---

## Author Response (AR2)

**Response to Referee #1**

After going through the revisions of "The effect of rainfall amount and timing on annual transpiration in a grazed savanna grassland", I find that the uncertainties and limitations to the study have still not been adequately addressed. In particular, estimation of ecosystem transpiration from all methods is difficult, and methods relying on eddy covariance methods, while improving in recent years, are still uncertain (Stoy et al 2019, Nelson et al 2020, Scott et al 2020, Hu and Lei 2021). In depth analysis from sites with many quality measurements and expert knowledge (such as what is presented in this manuscript) are vital to understanding both the true ecosystem transpiration dynamics, as well as uncertainties in the transpiration estimation methodologies. Therefore, utilization of multiple methods for estimating transpiration with different underlying assumptions is important to understand if the patters observed, such as the findings here that annual T and T/ET are linearly related to the early season precipitation, are robust.

We have provided the results for all three methods in the manuscript and in the supplement. The N18 method result is now included in fig. 7. In addition, the supplementary fig. 9, 10 and 11 show relation to all rainfall variables and EVI for all methods. Result section now reads:

*The annual T/ET and T were linearly related to early wet-season storm frequency for the B16 and Z16 methods (Fig. 7), while the relation of early season P, mid wet season P, annual P or mean annual EVI with annual T /ET, and T were more scattered (Fig. S9, S10, and S11). For the N18 method, the annual T was linearly related to mid wet season P (Fig. S10), and annual T and T/ET were linearly related to mean EVI (Fig. S11).*

The manuscript uses three ET partitioning methods, with the Berkelhammer and uWUE methods being very similar both in calculation and underlying assumptions. The third method (TEA), also shares many assumptions (particularly that T=ET during some periods), but is the most different of the three. The previous version dismissed the uWUE and TEA methods stating that " The T/ET values in the late wet season of 2015 based on the TEA and uWUE methods are likely overestimates, given the decrease in GPP and low EVI values during this drought year", and did not present findings based on the other methods in many of the results, making it difficult to understand the robustness of the findings and conclusions. I think it is very important to show all the results, even if in the supplementary materials, particularly Figure 7 which is the main finding. However, while all three methods are now presented in many of the plots, the TEA method is still excluded in the key findings.

We agree – not showing the results from all methods in Fig. 7 was an oversight. As mentioned above we have now included results for all methods. The water balance components for Z16 and N18 are already presented in Table S2 in the latest supplement.

The argumentation used for dismissing the TEA method is now based on a soil evaporation model, with the derived parameter of De for the Berkelhammer method being closes to those reported in other literature, particularly Hu and Lei, 2021. However, the Hu and Lei study also compared seven different ET partitioning methods (including the uWUE and TEA methods) and found that the TEA method performed the best, which is a direct contradiction of what is reported here.

The issue here is not which method is correct or incorrect, because all the partitioning methods are

all wrong in some way. I strongly advise the authors to revise the results here to take this uncertainty into account and understand how the differences in partitioning methods may impact the interpretation, or at the very least report the findings from all the methods.

As we have state above, we now provide the results from all the methods in this major revision.

Our analysis does not intend to dismiss the N18 method. Indeed, we have no intention of giving such impression as a reason for our analyses. We have provided all the results for the N18 method and we do give reasons why it is likely an overestimate at this site. Given that, we focus on what we consider more reliable estimates of transpiration to evaluate controls of this flux in our study site. Including N18 results in all discussion points would render the manuscript incomprehensible and will draw attention away from the novel results.

Overall, the reasons why the $T$ from N18 method is likely an overestimate at this site are: (1) The estimated soil desorptivity ($D_e$) values are much lower than what is reported in the literature or in the Hu and Lei (2021) study, meaning that evaporation is probably underestimated and transpiration overestimated after rain events at daily scale (Fig. 4 in the manuscript). (2) The range of annual ratio of transpiration to ET ($T_{N18}$/ET) is from 0.54 to 0.65 which is much higher range than from the other methods and higher than the reported range from the grassland in Arizona (Scott et al, 2021) and from the Nylsvley savanna with 30% tree cover (Scholes and Walker, 1993). (3) When using N18 to estimate total annual T, and subtracting the estimated grass transpiration, the difference which is an estimate of annual tree transpiration is 193 mm averaged over the three wet years. This estimate is much higher than reported in the literature at similar sites. Taken together, we consider this a strong evidence that N18 overestimate transpiration at this site. We have added this point to the discussion.

We prefer not to delve further into assessment of the potential causes for the N18 method to generate overestimation of $T$; this will be speculative, beyond the scope of our study, and will draw attention away from the focus of the study. We note, however, that the likely reason transpiration was not overestimated in Hu and Lei (2021) study is that their study site is an irrigated field that does not experience long periods of transpiration reducing drought.

Hu, Xingyu, and Huimin Lei. "Evapotranspiration Partitioning and Its Interannual Variability over a Winter Wheat-Summer Maize Rotation System in the North China Plain." Agricultural and Forest Meteorology 310 (November 2021): 108635. https://doi.org/10.1016/j.agrformet.2021.108635.

Nelson, Jacob A., Oscar Pérez-Priego, Sha Zhou, Rafael Poyatos, Yao Zhang, Peter D. Blanken, Teresa E. Gimeno, et al. "Ecosystem Transpiration and Evaporation: Insights from Three Water Flux Partitioning Methods across FLUXNET Sites." Global Change Biology, October 6, 2020. https://doi.org/10.1111/gcb.15314.

Scott, Russell L., John F. Knowles, Jacob A. Nelson, Pierre Gentine, Xi Li, Greg Barron-Gafford, Ross Bryant, and Joel A. Biederman. "Water Availability Impacts on Evapotranspiration Partitioning." Agricultural and Forest Meteorology, November 2020, 108251. https://doi.org/10.1016/j.agrformet.2020.108251.

Stoy, Paul C., Tarek S. El-Madany, Joshua B. Fisher, Pierre Gentine, Tobias Gerken, Stephen P. Good, Anne Klosterhalfen, et al. "Reviews and Syntheses: Turning the Challenges of Partitioning Ecosystem Evaporation and Transpiration into Opportunities." Biogeosciences 16, no. 19 (October 1, 2019): 3747–75. https://doi.org/10.5194/bg-16-3747-2019.

**Response to Referee #2**

This is my second review of the manuscript. The authors did a good job of addressing my concerns with the first version, and the paper reads much better now. Thank you for considering my comments and suggestions.

I have only a few small comments that you might want to address:

1. L22. What area? Have you introduced what area you're talking about?

Southern African drylands. We have rephrased the sentence.

2. P.6, L3-5. This is so great to see you considering this. So many studies don't.

Thank you for this comment.

3. P. 12 section 2.8. This model of grass T seems to have some assumptions that are worth justifying here. The first being that there is no drainage below 60 cm, the second being that E is constant which would underestimate E and overestimate grass T after a rain event and may underestimate grass T later into an interstorm period. I'm confused why a dynamic soil E was not used.

The algorithm does not explicitly estimate drainage. The main purpose of using the model instead of relying on difference of soil moisture values between layers is to allow quantification of the characteristic declines in soil moisture corresponding to RWU events. The periods when a certain soil moisture level reflects percolation are excluded by the algorithm. The soil moisture values at the bottom layer (100 cm deep) are constant during the measurement period, suggesting that drainage is small.

We did test the use of the dynamic $E$ from each model, but the EC estimated $E$'s are over 60% of grass $T$, which is unrealistically high amount. This is because the soil moisture measurements may not fully capture the soil evaporation of the EC footprint. The estimated grass $T$ is in general low immediately after large rain events due to the exclusion of percolation events. For these reasons, the constant $E$ was used, resulting in a reasonable estimate of annual grass $T$.

4. P.23, L13-15. What about early season or total growing season amounts as a predictor of T and T/ET? I don't recall you considering this, but in a water-limited system I would think the amount of rainfall would be an important control.

We have now added supplementary figures for relations between $T$ and $T$/ET and early wet season $P$, mid-wet season $P$, annual $P$ and mean annual EVI, for all methods. These predictors have more scattered relation with $T$ and $T$/ET for the B16 and Z16 methods. We do emphasize the location (highveld grasslands) because the result is supported by the other experimental study from a similar highveld grassland (Swemmer et al., 2007).